

# Estimating the timescale-dependent uncertainty of paleoclimate records—a spectral approach. Part I: Theoretical concept

Torben Kunz[1], Andrew M. Dolman[1], and Thomas Laepple[1,2]

[1]Alfred-Wegener-Institut Helmholtz-Zentrum für Polar- und Meeresforschung, Research Unit Potsdam, Telegrafenberg A45, 14473 Potsdam, Germany
[2]University of Bremen, MARUM – Center for Marine Environmental Sciences and Faculty of Geosciences, 28334 Bremen, Germany

**Correspondence:** Torben Kunz (torben.kunz@awi.de)

**Abstract.**

Proxy records represent an invaluable source of information for reconstructing past climatic variations, but they are associated with considerable uncertainties. For a systematic quantification of these reconstruction errors, however, knowledge is required not only of their individual sources but also of their auto-correlation structure, as this determines the timescale dependence of their magnitude, an issue that is often ignored until now. Here a spectral approach to uncertainty analysis is provided for paleoclimate reconstructions obtained from single sediment proxy records. The formulation in the spectral domain, rather than the time domain, allows for an explicit demonstration as well as quantification of the timescale dependence that is inherent in any proxy-based reconstruction uncertainty. This study is published in two parts.

In this first part, the theoretical concept is presented and analytic expressions are derived for the power spectral density of the reconstruction error of sediment proxy records. The underlying model takes into account the spectral structure of the climate signal, seasonal and orbital variations, bioturbation, sampling of a finite number of signal carriers, uncorrelated measurement noise, and it includes the effects of spectral aliasing and leakage. The uncertainty estimation method, based upon this model, is illustrated by simple examples. In the second part of this study, published separately, the method is implemented in an application-oriented context, and more detailed examples are presented.

## 1 Introduction

The central issues of climate sciences include the estimation, understanding and prediction of climatic variations, across ranges of space and timescales that are relevant to the specific field of study. From an inductive perspective, such studies are necessarily based on observational data which the variability may be estimated from, whereas from a deductive perspective observational data are needed in the course of validation of theories and models. For certain fields of study instrumental or satellite data may provide a useful data source. Nonetheless, once processes are studied that involve climate states or variations at times before the instrumental era, or that involve timescales longer than this, reconstructions obtained from paleoclimate proxies become indispensible. Such proxy records reveal imprints of past climatic conditions, created by, for example, impacts on the calcification of the shells of marine organisms (Nürnberg et al., 1996), now preserved in sea sediments, on terrestrial pollen





assemblages archived in lake sediments (Birks and Seppä, 2004), or on stable water isotopes that can be recovered from ice-
cores (Jouzel et al., 1997). Proxy-based reconstructions, however, are associated with notable uncertainties that are often much
larger than those of instrumental data (Münch and Laepple, 2018; Reschke et al., 2019), and which can emerge from a variety
of sources—they are essentially highly noisy and distorted observations of selected climate variables. Hence, an important task
of the paleoclimate research field is to provide thourough quantitative estimates of these reconstruction uncertainties.

Possible sources of reconstruction uncertainties include, but are not limited to, measurement errors occurring in the labora-
tory (Rosell-Melé et al., 2001; Greaves et al., 2008), errors induced by smoothing processes like bioturbation affecting sediment
archives (Berger and Heath, 1968; Goreau, 1980) or diffusion within ice-cores (Johnsen, 1977; Whillans and Grootes, 1985),
aliasing of variability from higher than the resolved frequencies (e.g., from ENSO or the seasonal cycle; see, for example,
Thirumalai et al., 2013; Laepple et al., 2018), proxy seasonality (Jonkers and Kučera, 2015), potentially interacting with mod-
ulations of the seasonal cycle amplitude caused by slow orbital variations (Huybers and Wunsch, 2003; Laepple et al., 2011),
uncertainties in the understanding of the climate-proxy relationship (including calibration errors; Tierney and Tingley, 2014),
and others, depending on the type of proxy used.

It turns out that a careful and systematic investigation of these reconstruction uncertainties is indispensible, if we are to
properly exploit the source of information contained in proxy archives, for such important issues like the estimation of the
future evolution of natural and forced climate variability. Until now, however, reconstruction uncertainty estimates often lack
the required accuracy (Lohmann et al., 2013; Reschke et al., 2019). In particular, one issue that deserves more detailed consid-
eration is the timescale dependence of the reconstruction uncertainties (Amrhein, 2019). Although some of their sources like
measurement errors will often be independent and, thus, uncorrelated between individual measurements, others like smoothing
processes and orbital variations, in conjunction with proxy seasonality, have the potential to create serially correlated uncer-
tainties (i.e., they are auto-correlated in the time domain). Thus, some uncertainty components may be described by white
noise, while others may have the properties of red noise or an even more complex auto-correlation structure. The direct, and
practically relevant, implication of this is the fact that, when averaging the proxy-based climate reconstruction over some time
interval (e.g., by applying a moving average filter), the uncertainties may shrink at a different rate than if they were purely
white noise.

One possibility to estimate the auto-correlation structure of reconstruction uncertainties consists in the application of proxy
forward models that generate proxy time series from climate (model) time series (see, for example, Evans et al., 2013; Dee
et al., 2015; Dolman and Laepple, 2018). Specifically, the auto-correlation structure may then be inferred from ensembles of
such simulated proxy time series. This approach is flexible regarding the complexity of the uncertainty-generating processes
included in the model, but the insights gained from its application are limited by the fact that it represents a try-and-error
strategy. Moreover, the involved numerical simulations easily become computationally expensive. Therefore, it is useful and
desirable to complement this by an alternative approach that allows for a systematic understanding of the auto-correlation
structure of the reconstruction error components from an analytic point of view.

Accordingly, the aim of this paper is to provide a conceptual approach and, based thereon, an analytically derived method
to estimate timescale-dependent reconstruction uncertainties, for the example of sediment archives. Specifically, the method





yields uncertainty estimates, given a set of parameters that specify (i) the spectral structure of a supposed true climate signal, (ii) seasonal and orbital variations, (iii) proxy seasonality, (iv) bioturbation, (v) archive sampling parameters, (vi) sampling of a finite number of signal carriers, (vii) uncorrelated measurement noise, and it takes into account the effects of spectral aliasing and leakage. The representation of smoothing by bioturbation limits the validity of the method in its current form to proxy archives from sea and lake sediments. However, it has the potential to be generalized to other sedimentary archives such as ice-cores, by modifying the smoothing operator to represent isotopic diffusion.

The pivotal idea of our approach to address the timescale dependence of the uncertainty consists in the derivation of its power spectrum, as the spectrum is directly related (by the Wiener-Khintchine theorem, see Priestley, 1981) to the auto-correlation structure which, in turn, determines how the uncertainty scales with timescale (e.g., the length of an averaging interval). The convenience of the obtained mathematical expressions for the uncertainty power spectrum is twofold: (a) They can be used to acquire a qualitative understanding of the effects and relative importance of the various sources of uncertainty. (b) They can serve to obtain quantitative uncertainty estimates for specific practical applications in paleoclimate science.

Part I of this study provides the theoretical basis of the uncertainty estimation method. In section 2 the underlying reconstruction uncertainty model is defined in the time domain. Section 3 translates the model into the spectral domain by deriving the corresponding uncertainty power spectrum. Section 4 summarizes the results and demonstrates how timescale-dependent uncertainties can be obtained from the spectrum. The method and its limitations are discussed in section 5, followed by the final conclusions in section 6. Part II of this study, published separately (see Dolman et al., 2019), demonstrates the practical applicability of the method and also provides a software implementation for practical uncertainty estimation purposes, the so-called Proxy Spectral Error Model (PSEM).

## 2  Reconstruction uncertainty model

Before we can formulate our timescale-dependent uncertainty estimation method, we have to provide a precise definition of the underlying reconstruction uncertainty model, including our assumptions and simplifications that allow for an analytic treatment of the problem. Specifically, in order to define the uncertainty model, we need to

- suppose a structure of the true climate signal, which the final uncertainty estimates will be based upon, because some uncertainty components and their timescale dependence are subject to that structure

- make simplifying assumptions regarding the archive formation, concerning proxy seasonality, the climate-proxy relationship, the sediment accumulation rate and the effects of bioturbation mixing

- specify the archive sampling and measurement procedure

- define the reconstruction error as the difference between the obtained climate reconstruction and a suitable reference climate

- define the reconstruction uncertainty in terms of the expected value of the squared reconstruction error.





**Table 1.** Parameters of the reconstruction uncertainty model, as defined in section 2.

| Parameter | Symbol |
| --- | --- |
| Seasonal cycle variance | $\sigma_c^2$ |
| Seasonal cycle frequency | $\nu_c$ |
| Expected seasonal cycle phase | $\langle \phi_c \rangle_{\phi_c}$ |
| Seasonal phase uncertainty | $\Delta_{\phi_c}$ |
| Amplitude modulation variance | $\sigma_a^2$ |
| Amplitude modulation frequency | $\nu_a$ |
| Amplitude modulation phase | $\phi_a$ |
| Proxy abundance timescale | $\tau_p$ |
| Bioturbation timescale | $\tau_b$ |
| Sediment sampling timescale | $\tau_s$ |
| Sampling interval | $\Delta t$ |
| Length of proxy record | $T$ |
| Number of signal carriers | $N$ |
| Measurement error variance | $\sigma_\mu^2$ |
| Reference climate averaging timescale | $\tau_r$ |

Accordingly, the reconstruction uncertainty model can be thought of, conceptually, as an operator that takes as its arguments the supposed structure of the true climate signal and a set of parameters that appear in the mathematical formulation of the above assumptions. The remainder of this section is concerned with the details of the above five steps, including an explanation of the involved parameters. A complete list of the model parameters is provided by Table 1. Note, that the reconstruction uncertainty model defined in this section is closely related to the proxy forward model of Dolman and Laepple (2018).

## 2.1 Climate signal

We assume that the supposed true climate signal consists of two components: a stochastic signal $X(t)$, that represents the signal to be reconstructed from the proxy record, and a deterministic signal $Y(t)$, that represents the seasonal cycle, the amplitude of which is modulated by slow orbital variations. In addition, we make the simplifying assumption that $X(t)$ and $Y(t)$ are stochastically independent.

The stochastic signal $X(t)$ is modelled as a zero-mean stochastically continuous stationary random process, with infinite and continuous time parameter $t$, and that has a purely continuous power spectrum (i.e., the spectrum has no discrete components). The actual structure of $X(t)$ is to be specified in the spectral domain (see section 3), and the uncertainty estimation method is constructed such that any spectral structure can be specified as long as it is consistent with the abovementioned properties of $X(t)$. An illustration of one realization of such a random process is given by the gray-red line in Fig. 1b (obtained from a





surrogate time series that obeys a simple power-law frequency scaling). It can be thought of as a time series of anomalies of a climate signal after removal of the climatological seasonal cycle.

The deterministic signal $Y(t)$ is modelled as a single harmonic oscillation, that represents a simplified seasonal cycle, which is amplitude modulated by another single harmonic oscillation with a much longer period. Thus, $Y(t)$ has a purely discrete power spectrum. Such a deterministic signal can be written as

$$Y(t) = \sigma_c\sqrt{2}\cos(\phi_c + 2\pi\nu_c t)\left[1 + \sigma_a\sqrt{2}\cos(\phi_a + 2\pi\nu_a t)\right], \tag{1}$$

where $\nu_c = (1\,\mathrm{yr})^{-1}$ and $\nu_a \ll (1\,\mathrm{yr})^{-1}$ are the frequencies of the seasonal cycle and of its slow amplitude modulation, respectively, $\sigma_c^2$ and $\sigma_a^2$ are the corresponding variances of those oscillations, and $\phi_c$ and $\phi_a$ are their phases. The square bracket term represents the amplitude modulation factor that specifies the time-varying envelope of the seasonal cycle oscillation. Note, that $\sigma_c$ has the same units as $X(t)$ and $Y(t)$, whereas $\sigma_a$ is dimensionless, as it determines the fraction by which the amplitude of the seasonal cycle varies. In particular, $\sigma_c\sqrt{2}$ is the half-amplitude of the unmodulated seasonal cycle, and $\sigma_a\sqrt{2}$ is the fraction by which the seasonal cycle amplitude changes over an orbital modulation cycle. Furthermore, it is required that $\sigma_a\sqrt{2} < 1$, or equaivalently $\sigma_a^2 < 1/2$, to avoid flipping seasons by a negative amplitude modulation factor (which would correspond to unrealistically strong effects of orbital variations). The deterministic signal is illustrated by the gray-red line in Fig. 1c. Note, that only for the purpose of illustration the modulation frequency has been set to $\nu_a = (130\,\mathrm{yr})^{-1}$ in this figure, although a realistic value would be $\nu_a = (23\,\mathrm{kyr})^{-1}$, for example, if it were to represent an idealized planetary precession cycle.

## 2.2 Archive formation

To reflect proxy seasonality, we assume a seasonally confined time window during which the proxy is abundant. Thus, the climate signal, and, in particular, the seasonal cycle, is recorded only during those seasons. The length of this proxy abundance window is specified by the parameter $\tau_p$, and the timing of the center of this window with respect to the seasonal cycle is specified by $\phi_c$, as it appears in (1). Accordingly, if $\phi_c = 0$, then the abundance window is centered at the maximum of the seasonal cycle (i.e., the summer season, if the climate signal $X(t) + Y(t)$ represents temperature, for example), setting $\phi_c = \pm\pi/2$ centers the window at either of the zero-crossings (spring or autumn), $\phi_c = \pi$ at the minimum (winter), and likewise for all other phases. The seasonality parameters are required to fulfill the relations $\tau_p \leq 1\,\mathrm{yr}$ and $-\pi < \phi_c \leq \pi$. If $\tau_p = 1\,\mathrm{yr}$, there is no seasonality and the parameter $\phi_c$ has no effect. Since in this formulation $\tau_p$ and $\phi_c$ are fixed, the above assumptions imply that we are neglecting any changes of proxy seasonality, caused by, for example, habitat tracking. Specifically, there is no adaptation of proxy seasonality to changes in the seasonal cycle amplitude, nor to variations of the stochastic component of the climate signal at any timescales. The effect of proxy seasonality defined in this way is illustrated, in Figs. 1b and c, by the red line segments, highlighting that part of the signal that is recorded by the proxy. In this example the proxy abundance window is set to cover the seasons around the maximum of the seasonal cycle.

In the following we also neglect any uncertainties regarding the climate-proxy relationship, including calibration errors. Furthermore, we assume a known and constant sediment accumulation rate. Thus, we will treat all signals simply as a function of time, and assume that the constant depth-time relationship is given as an independent information.



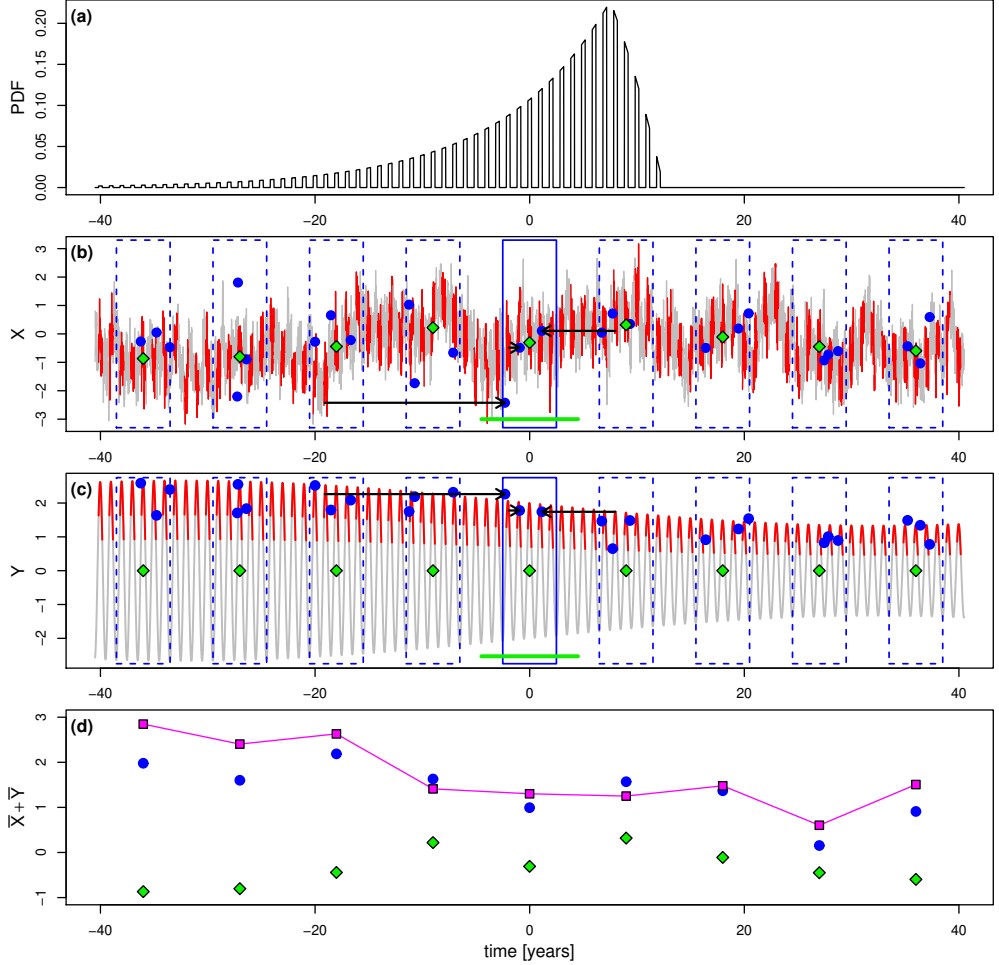

**Figure 1.** Schematic illustration of the reconstruction uncertainty model. (a) Probability density function (as a function of time lag) that describes the combined effects of bioturbation (with timescale $\tau_b$), of sediment sample thickness, and of proxy seasonality. (b) Stochastic component $X(t)$ of the climate signal, gray line, with dates highlighted in red that fall into the proxy abundance window (of length $\tau_p$), to reflect proxy seasonality. Blue rectangles indicate time intervals of length $\tau_s$, covered by the sediment slices. From each slice a finite number of signal carriers ($N = 3$ in this example) are retrieved from random positions within the slice, indicated by the blue dots, each of which carries the signal from the time at which it settled down on the surface of the sediment, before it was mixed to its current position by bioturbation (indicated by black arrows for the central slice). Green squares indicate the reference climate signal, obtained by averaging $X(t)$ over intervals of length $\tau_r$, indicated by the green line for the central point. (c) Same as (b), but for the deterministic component $Y(t)$, that represents the amplitude modulated seasonal cycle. (d) Total reconstructed signal (blue), obtained by averaging over the $N$ signal carriers from each slice, reference climate signal (green), and the difference between them (magenta) that represents the reconstruction error; at a sampling interval $\Delta t$. Measurement errors are neglected in this illustration. timescale parameters are set to $\tau_b = 10$ yr, $\tau_s = 5$ yr, $\tau_r = 9$ yr, $\Delta t = 9$ yr, $\tau_p = 1/3$ yr.





Signal smoothing by sediment mixing caused by bioturbation is assumed to occur instantly and uniformly within the upper-most layer of the sediment. The thickness of this layer, the bioturbation depth, can be divided by the sediment accumulation

rate to obtain the corresponding bioturbation timescale $\tau_b$. Under the aforementioned assumptions the effects of bioturbation can be described by a probability density function (PDF) of the form (Berger and Heath, 1968)

$$f_b(\epsilon) = \begin{cases} \tau_b{}^{-1}\exp[(\epsilon - \tau_b)/\tau_b] & \text{if } \epsilon \leq \tau_b \\ 0 & \text{otherwise} \end{cases}, \tag{2}$$

where $\epsilon$ has units of time. This PDF specifies at which probability a single signal carrier, retrieved from the archive at position $t = t_0$, has settled down on the surface of the sediment and, thus, has recorded the climate signal at a given time $t = t_0 + \epsilon$.

Essentially, it states that a signal carrier retrieved at $t = t_0$ cannot have its origin at times later than $t = t_0 + \tau_b$, but that it can have its origin arbitrarily far in the past relative to $t_0$, although with exponentially decreasing probability. Thus, $\epsilon$ can be interpreted as the timing error, caused by bioturbation, that is associated with the signal recorded by an individual signal carrier. In Figs. 1b and c, the effect of bioturbation is illustrated by the black arrows, indicating the net mixing paths (i.e., the timing errors $\epsilon$) of three selected signal carriers (blue dots).

## 2.3 Sampling and measurement procedure

We assume that the archive is sampled by taking slices of sediment, the thickness of which corresponds to time intervals of length $\tau_s$, and which are taken at distances (measured from center to center) corresponding to a sampling interval $\Delta t$. This sampling procedure is illustrated in Figs. 1b and c by the blue rectangles, indicating individual sediment slices. The total length of the record is denoted by $T$. For mathematical reasons that become apparent in section 3, it is required that $T$ is a multiple of

$\Delta t$, and that $\Delta t$ and $\tau_s$ are multiples of 1 year. Setting $\tau_s < \Delta t$ corresponds to discontinuous sampling, $\tau_s = \Delta t$ to continuous sampling, and $\tau_s > \Delta t$ to sampling with overlap. The effects of these cases are discussed by Amrhein (2019).

Because signal carriers are retrieved from arbitrary positions within each slice, the effect of the sediment sample timescale $\tau_s$ can be described by convolving the bioturbation PDF, $f_b(\epsilon)$, with a slice PDF that has the shape of a moving average window, $f_s(\epsilon) = \tau_s^{-1}\Pi(\epsilon; \tau_s)$, and which is essentially blurring the edge of $f_b(\epsilon)$; where the symbol $\Pi(t; \tau)$ denotes the rectangle

function

$$\Pi(t; \tau) = \begin{cases} 1 & \text{if } |t| \leq \tau/2 \\ 0 & \text{otherwise} \end{cases}. \tag{3}$$

Thus, the PDF of the timing errors, which describes the combined effects of bioturbation and of sampling slices of sediment, can be written as

$$f_{bs}(\epsilon) = f_s(\epsilon) * f_b(\epsilon), \tag{4}$$

Hence, if there were no bioturbation ($\tau_b \rightarrow 0$) and if single signal carriers were retrieved from infinitesimally thin slices ($\tau_s \rightarrow 0$), then this PDF would reduce to a Dirac delta function, $f_{bs}(\epsilon) \rightarrow \delta(\epsilon)$, in which case the above sampling procedure would





yield the discrete climate signal $X_n + Y_n = X(t_n) + Y(t_n)$, where $t_n = n\Delta t$ (with $n = 0, \pm 1, \pm 2, \ldots$). In the general case with $\tau_b > 0$ and $\tau_s > 0$, we can express the result of the sampling procedure as a discrete signal with jittered sampling,

$$X_n^{(j)} + Y_n^{(j)} = X(t_n + \epsilon_n^{(j)}) + Y(t_n + \epsilon_n^{(j)}), \tag{5}$$

with $\epsilon_n^{(j)} \sim f_{bs}(\epsilon)$, where $\epsilon_n^{(j)}$ represents the sampling jitter and $f_{bs}(\epsilon)$ the jitter PDF. In the above terminology, $\epsilon_n^{(j)}$ represents the timing error of a single signal carrier retrieved from a slice centered at $t = t_n$.

Finally, we need to include the effect of proxy seasonsality as defined in the previous subsection. This is accomplished through multiplying $f_{bs}(\epsilon)$ by a proxy seasonality function $p(\epsilon)$, that is given by the convolution of the Dirac comb function $\text{III}(\epsilon; \nu_c^{-1})$ with the rectangle function $(\tau_p \nu_c)^{-1}\Pi(\epsilon; \tau_p)$, that is,

$$p(\epsilon) = (\tau_p \nu_c)^{-1}\Pi(\epsilon; \tau_p) * \text{III}(\epsilon; \nu_c^{-1}); \tag{6}$$

where the Dirac comb function $\text{III}(t; \tau)$ is defined as a series of Dirac delta functions $\delta(t)$,

$$\text{III}(t; \tau) = \sum_{k=-\infty}^{\infty} \delta(t - k\tau). \tag{7}$$

It turns out that in the limit of vanishing proxy seasonality ($\tau_p \to 1$ yr), the proxy seasonality function becomes constantly one, $p(\epsilon) \to 1$, whereas in the limit of maximum proxy seasonality ($\tau_p \to 0$), the proxy seasonality function reduces to the Dirac

comb function, $p(\epsilon) \to \text{III}(\epsilon; \nu_c^{-1})$. From the above, the discrete climate signal, obtained from the sampling procedure, may still be written as in (5), but with the sampling jitter

$$\epsilon_n^{(j)} \sim p(\epsilon) f_{bs}(\epsilon) \tag{8}$$

now being drawn from the full jitter PDF, $p(\epsilon) f_{bs}(\epsilon)$, which describes the combined effects of bioturbation, of sampling slices of sediment, and of proxy seasonality. The proof that the full jitter PDF defined in this way integrates to unity follows in

section 3. The structure of the full jitter PDF is illustrated by Fig. 1a.

In practice a finite number $N \geq 1$ of signal carriers is retrieved from each sediment slice, rather than just a single signal carrier, and subsequently a single proxy measurement is performed in the laboratory on the collection of those $N$ signal carriers, representing an average proxy value. This can be expressed as

$$\bar{X}_n + \bar{Y}_n = \frac{1}{N} \sum_{j=1}^{N} \left[ X_n^{(j)} + Y_n^{(j)} \right]. \tag{9}$$

In addition, we assume that the involved sampling jitter, $\epsilon_n^{(j)}$, is uncorrelated (i.e., white) in terms of both, $n$ and $j$,

$$\text{Cor}\left(\epsilon_n^{(j)}, \epsilon_{n'}^{(j')}\right) = 0 \qquad \text{if} \quad n \neq n' \quad \text{or} \quad j \neq j', \tag{10}$$

which reflects our assumption that the bioturbation mixing paths of the individual signal carriers within the sediment do not affect each other. Furthermore, it is required that $\epsilon_n^{(j)}$ is a stationary process, to reflect our assumptions of a fixed bioturbation





depth and a constant sediment accumulation rate. Finally, we take $\epsilon_n^{(j)}$ as independent of $X(t)$ and $Y(t)$, corresponding to the

assumption that bioturbation does not depend on the climate.

In general, each laboratory measurement is associated with a measurement error $\mu_n$, the magnitude of which may be characterized in terms of its variance $\sigma_\mu^2$. Thus, the final reconstruction time series is given by $\bar{X}_n + \bar{Y}_n + \mu_n$, although we will omit $\mu_n$ in the following as it is assumed to be white noise and, thus, it can easily be added at the very end of the entire uncertainty estimation procedure.

## 2.4   Definition of reconstruction error

The reconstruction error can now be defined as the difference between the obtained climate reconstruction (9) and a suitable reference climate

$$\tilde{X}_n + \tilde{Y}_n = \tilde{X}(t_n) + \tilde{Y}(t_n), \tag{11}$$

where

$$\tilde{X}(t) + \tilde{Y}(t) = \tau_r^{-1} \Pi(t; \tau_r) * [X(t) + Y(t)] \tag{12}$$

is the supposed true climate signal smoothed with a moving average filter with timescale $\tau_r$, which is then subsampled at the same discrete times $t_n$. Here we require that $\tau_r$ is a multiple of 1 year, such that $\tilde{Y}_n = 0$ because it is then an average over a number of complete seasonal cycles. Thus, we obtain the reconstruction error time series as

$$E_n = E_{X,n} + E_{Y,n}, \tag{13}$$

with the error components

$$E_{X,n} = \bar{X}_n - \tilde{X}_n \qquad \text{and} \qquad E_{Y,n} = \bar{Y}_n. \tag{14}$$

An example of one realization of the discrete climate reconstruction (9), reference climate (11), and reconstruction error time series (13) is illustrated by Fig. 1d.

## 2.5   Reconstruction error versus uncertainty

The so defined reconstruction error $E_n$ refers only to a single realization of the stochastic processes $X$ and $\epsilon$, which are specified in terms of their power spectral density and PDF, respectively. Thus, to obtain a suitable measure of the reconstruction uncertainty that characterizes the magnitude of possible errors, under the specified stochastic properties of $X$ and $\epsilon$, we define the root-mean-square (RMS) reconstruction error $\mathcal{E}_n$ by

$$\mathcal{E}_n^2 = \left\langle \left\langle E_n^2 \right\rangle_X \right\rangle_\epsilon, \tag{15}$$

where $\langle \cdot \rangle_X$ and $\langle \cdot \rangle_\epsilon$ denote the expected value operators with respect to $X$ and $\epsilon$, respectively. Then substitution from (13) yields

$$\mathcal{E}_n^2 = \left\langle \left\langle (E_{X,n} + E_{Y,n})^2 \right\rangle_X \right\rangle_\epsilon = \left\langle \left\langle E_{X,n}^2 \right\rangle_X \right\rangle_\epsilon + \left\langle E_{Y,n}^2 \right\rangle_\epsilon, \tag{16}$$





because $X$ and $Y$ are assumed to be independent, and because $X$ is a zero-mean process, that is, $\langle X \rangle_X = 0$ and, thus, $\langle\langle E_{X,n} \rangle_X \rangle_\epsilon = 0$.

As will be shown in section 3, $E_{X,n}$ can be decomposed into two uncorrelated zero-mean stationary components as $E_{X,n} = F_{X,n} + W_{X,n}$, such that $F_{X,n}$ can be expressed as the result obtained by bandpass filtering the signal $X(t)$ in time, and then subsampling it at the discrete times $t_n$, and $W_{X,n}$ is a white noise process. Furthermore, it will be shown that $E_{Y,n}$ can be decomposed into two uncorrelated and generally non-stationary components as $E_{Y,n} = F_{Y,n} + W_{Y,n}$, such that $F_{Y,n}$ can be expressed as the result obtained by filtering and then subsampling the signal $Y(t)$, and $W_{Y,n}$ is a zero-mean white noise

process. Thus, we can write

$$\mathcal{E}_n^2 = \left\langle \left\langle F_{X,n}^2 \right\rangle_X \right\rangle_\epsilon + \left\langle \left\langle W_{X,n}^2 \right\rangle_X \right\rangle_\epsilon + F_{Y,n}^2 + \left\langle W_{Y,n}^2 \right\rangle_\epsilon, \tag{17}$$

where $F_{Y,n}$ is a deterministic signal.

In addition to the uncertainty caused by the stochasticity of $X$ and $\epsilon$, we can in principle equip any of the model parameters with its own uncertainty, and investigate how this contributes to the obtained reconstruction uncertainty. In the following we

apply this procedure to the seasonal phase $\phi_c$, as the seasonal timing of the proxy abundance is often a poorly constrained parameter. For this purpose, we need to specify a corresponding PDF of the seasonal phase. For simplicity, we choose the uniform PDF

$$f_{\phi_c}(\phi_c) = \Delta_{\phi_c}^{-1} \Pi(\phi_c - \langle \phi_c \rangle_{\phi_c}; \Delta_{\phi_c}), \tag{18}$$

with the seasonal phase uncertainty $0 \leq \Delta_{\phi_c} \leq 2\pi$. Note, that setting $\Delta_{\phi_c} = 2\pi$ does not imply vanishing proxy seasonality (as

this is expressed by setting $\tau_p = 1$ yr), but merely means that the seasonal timing of the proxy abundance window is completely unknown. The model parameters to be specified are now $\langle \phi_c \rangle_{\phi_c}$ and $\Delta_{\phi_c}$ (see also Table 1), rather than the single parameter $\phi_c$ which is treated as unknown according to $f_{\phi_c}(\phi_c)$. Now, to include the effect on the reconstruction uncertainty, we redefine the RMS reconstruction error $\mathcal{E}_n$ by applying the additional expected value operator $\langle \cdot \rangle_{\phi_c}$, with respect to $\phi_c$, to the right-hand side of (15), or likewise (17). Since $X$ does not depend on $\phi_c$, we obtain

$$\mathcal{E}_n^2 = \left\langle \left\langle \left\langle E_n^2 \right\rangle_X \right\rangle_\epsilon \right\rangle_{\phi_c} \tag{19}$$

$$= \left\langle \left\langle F_{X,n}^2 \right\rangle_X \right\rangle_\epsilon + \left\langle \left\langle W_{X,n}^2 \right\rangle_X \right\rangle_\epsilon + \left\langle F_{Y,n}^2 \right\rangle_{\phi_c} + \left\langle \left\langle W_{Y,n}^2 \right\rangle_\epsilon \right\rangle_{\phi_c}. \tag{20}$$

Hence, by noting that $\langle F_{Y,n}^2 \rangle_{\phi_c} = \langle F_{Y,n} \rangle_{\phi_c}^2 + \mathrm{Var}_{(\phi_c)}(F_{Y,n})$, we can finally write

$$\mathcal{E}_n^2 = \mathcal{B}_n^2 + \mathcal{U}_n^2, \tag{21}$$

with the squared reconstruction bias

$$\mathcal{B}_n^2 = \left\langle F_{Y,n} \right\rangle_{\phi_c}^2, \tag{22}$$

and the squared reconstruction uncertainty

$$\mathcal{U}_n^2 = \mathcal{U}_{(1)}^2 + \mathcal{U}_{(2)}^2 + \mathcal{U}_{(3),n}^2 + \mathcal{U}_{(4),n}^2, \tag{23}$$





the components of which are given by

$$\mathcal{U}_{(1)}^2 = \text{Var}_{(X,\epsilon)}(F_{X,n}), \qquad \mathcal{U}_{(2)}^2 = \text{Var}_{(X,\epsilon)}(W_{X,n}), \qquad \mathcal{U}_{(3),n}^2 = \text{Var}_{(\phi_c)}(F_{Y,n}), \qquad \mathcal{U}_{(4),n}^2 = \big\langle \text{Var}_{(\epsilon)}(W_{Y,n}) \big\rangle_{\phi_c}. \qquad (24)$$

Note, that from $\mathcal{U}_{(1)}^2$ and $\mathcal{U}_{(2)}^2$ the time index $n$ has been dropped to indicate the stationarity of these uncertainty components. It turns out that $\mathcal{E}_n^2$ represents the expected power of the reconstruction error at a given time $t_n$, which, according to (21), is decomposed into the power $\mathcal{B}_n^2$ contained in the reconstruction bias, and the variance $\mathcal{U}_n^2$ that quantifies the scatter around the bias.

   The individual components are to be interpreted as follows: The component $\mathcal{U}_{(1)}$ quantifies the reconstruction uncertainty
that arises from the difference between (i) the smoothing effect on $X(t)$ caused by bioturbation and by sampling from slices of sediment, and (ii) the smoothing effect on $X(t)$ caused by the moving average window used to obtain the reference climate. Since the two smoothing effects represent low-pass filters with different cut-off frequencies, they act together as a bandpass filter on $X(t)$ (as shown by Amrhein, 2019). This uncertainty component represents the total smoothing effect in the limit of infinitely many signal carriers being retrieved from each slice of sediment ($N \to \infty$). If only a finite number of signal carriers
is retrieved from each slice, there is an additional residual that is not averaged out in this case. This residual is quantified by the component $\mathcal{U}_{(2)}$. Likewise, the component $\mathcal{U}_{(4),n}$ quantifies the additional residual that arises from sampling only a finite number of signal carriers, but now pertaining to the deterministic signal $Y(t)$. This residual component also depends on the timing uncertainty of the seasonal proxy abundance, as specified by (18), because of the non-linear relation between the variance, aliased from the seasonal cycle, and the seasonal timing. Because the seasonal cycle amplitude is modulated over
time by orbital variations, this uncertainty component is non-stationary. On the other hand, in the limit of infinitely many signal carriers, the smoothing effects on $Y(t)$ leave nothing but a deterministic bias that obtains its only uncertainty, quantified by $\mathcal{U}_{(3),n}$, from the seasonal timing uncertainty. Finally, when averaging this bias across all possible seasonal timings, that are allowed according to (18), a purely deterministic error component is obtained which is quantified by the reconstruction bias $\mathcal{B}_n$. Further clarification of these interpretations will emerge from section 3.

Now, in order to formulate our timescale-dependent uncertainty estimation method, we need a spectral representation of the expected power $\mathcal{E}_n^2$. We will achieve this by deriving the power spectral density of the reconstruction error $E_n$, separately for its individual components, to obtain spectral representations of the squared reconstruction uncertainty components $\mathcal{U}_{(1)}^2$, $\mathcal{U}_{(2)}^2$, $\mathcal{U}_{(3),n}^2$, $\mathcal{U}_{(4),n}^2$, and of the squared reconstruction bias $\mathcal{B}_n^2$. This task is addressed in the following section. The reader who does not intend to follow the entire derivation may proceed directly with section 4 which summarizes the main results of section 3,
and then illustrates the method, based thereon, for estimating timescale-dependent reconstruction uncertainties.

## 3 Spectral representation of reconstruction uncertainty

The reconstruction uncertainty model, defined in the previous section, is now translated into the spectral domain. Since the two components of the supposed true climate signal have different properties, in the sense that $X(t)$ is a stationary random process with a continuous power spectrum, whereas $Y(t)$ is an non-stationary deterministic signal with a discrete power spectrum,



the two components require separate mathematical treatment. Accordingly, the derivation of the power spectral density of the reconstruction error $E_n$ is accomplished separately, in the following two subsections, for the components of $E_{X,n}$ that are based on the stochastic signal $X(t)$, and for those of $E_{Y,n}$ that are based on the deterministic signal $Y(t)$.

### 3.1 Stochastic signal: Continuous climate spectrum

A spectral representation of the stochastic signal component $X(t)$, with infinite and continuous time parameter $t$, is given by the Riemann-Stieltjes integral (see Priestley, 1981, section 4.11)

$$X(t) = \int_{-\infty}^{\infty} e^{i2\pi\nu t} dZ(\nu), \tag{25}$$

where $Z(\nu)$ is a complex-valued stochastic process, such that the power spectral density of $X(t)$ is given by

$$S_X(\nu) = \left\langle |dZ(\nu)|^2 \right\rangle_X / d\nu, \tag{26}$$

and where the $dZ(\nu)$ are zero-mean, orthogonal increments. Note, that a conventional Fourier representation of $X(t)$ does not exist because of the stochastic nature of the signal, and that it is $dZ(\nu)/d\nu$, rather than $Z(\nu)$, which formally plays the role of the Fourier transform in the above representation (Priestley, 1981). Likewise, the signal $X_n = X(t_n)$, sampled at the discrete times $t_n = n\Delta t$, has the spectral representation

$$X_n = \int_{-\infty}^{\infty} e^{i2\pi\nu t_n} dZ(\nu), \tag{27}$$

and the signal with jittered sampling, $X_n^{(j)} = X(t_n + \epsilon_n^{(j)})$, can be expressed as (Moore and Thomson, 1991)

$$X_n^{(j)} = \int_{-\infty}^{\infty} e^{i2\pi\nu(t_n + \epsilon_n^{(j)})} dZ(\nu). \tag{28}$$

Following the approach of Balakrishnan (1962), we consider its auto-covariance function $\left\langle \left\langle X_n^{(j)\star} X_{n'}^{(j)} \right\rangle_X \right\rangle_\epsilon$, where $(\cdot)^\star$ denotes the complex conjugate. By substitution from (28), and expressing the product of integrals as a double integral, we obtain (see Priestley, 1981, pp. 249–250, where the same is shown for the case without sampling jitter)

$$\left\langle \left\langle X_n^{(j)\star} X_{n'}^{(j)} \right\rangle_X \right\rangle_\epsilon = \left\langle \left\langle \int_{-\infty}^{\infty} \int_{-\infty}^{\infty} e^{-i2\pi\nu(t_n + \epsilon_n^{(j)})} e^{i2\pi\nu'(t_{n'} + \epsilon_{n'}^{(j)})} dZ^\star(\nu) dZ(\nu') \right\rangle_X \right\rangle_\epsilon \tag{29}$$

$$= \int_{-\infty}^{\infty} \int_{-\infty}^{\infty} e^{i2\pi t_n(\nu' - \nu)} e^{i2\pi\nu'(t_{n'} - t_n)} \left\langle e^{i2\pi(\nu'\epsilon_{n'}^{(j)} - \nu\epsilon_n^{(j)})} \right\rangle_\epsilon \left\langle dZ^\star(\nu) dZ(\nu') \right\rangle_X, \tag{30}$$

where we have used the independence of $\epsilon_n^{(j)}$ and $X(t)$. Now, from the orthogonality of the $dZ(\nu)$, it follows that $\left\langle dZ^\star(\nu) dZ(\nu') \right\rangle_X = 0$ whenever $\nu \neq \nu'$. Thus, the contribution to the integral (30) is non-zero only for $\nu = \nu'$, and the auto-covariance function can





then be expressed by the single integral [also using (26)]

$$
\left\langle \left\langle X_n^{(j)\star} X_{n'}^{(j)} \right\rangle_X \right\rangle_\epsilon = \int_{-\infty}^{\infty} e^{i2\pi\nu(t_{n'}-t_n)} C_{n,n'}(-\nu,\nu) S_X(\nu) d\nu,
\tag{31}
$$

with the characteristic function

$$
C_{n,n'}(\nu_1,\nu_2) = \left\langle e^{i2\pi(\nu_1 \epsilon_n^{(j)} + \nu_2 \epsilon_{n'}^{(j)})} \right\rangle_\epsilon.
\tag{32}
$$

Note, that without sampling jitter (i.e., with $\epsilon_n^{(j)} = 0$), expression (31) reduces to the Wiener-Khintchine theorem (see Priestley, 1981, for example), which states that the auto-covariance function of a signal and its power spectral density are a Fourier transform pair. Because $\epsilon_n^{(j)}$ is white, we have $\left\langle e^{i2\pi\nu(\epsilon_{n'}^{(j)} - \epsilon_n^{(j)})} \right\rangle_\epsilon = \left\langle e^{i2\pi\nu\epsilon_{n'}^{(j)}} \right\rangle_\epsilon \left\langle e^{-i2\pi\nu\epsilon_n^{(j)}} \right\rangle_\epsilon$ if $n \neq n'$ and, thus,

$$
C_{n,n'}(-\nu,\nu) = \begin{cases} 1 & \text{if } n = n' \\ |C(\nu)|^2 & \text{if } n \neq n' \end{cases},
\tag{33}
$$

where

$$
C(\nu) = \left\langle e^{i2\pi\nu\epsilon_n^{(j)}} \right\rangle_\epsilon
\tag{34}
$$

is the characteristic function (or the complex conjugate of the Fourier transform) of the jitter PDF, $p(\epsilon) f_{bs}(\epsilon)$, since using the definition of the expected value yields

$$
C(\nu) = \int_{-\infty}^{\infty} e^{i2\pi\nu\epsilon} p(\epsilon) f_{bs}(\epsilon) d\epsilon
\tag{35}
$$

$$
= \hat{p}(\nu) * \hat{f}_{bs}^{\star}(\nu)
\tag{36}
$$

$$
= \sum_{k=-\infty}^{\infty} \text{sinc}(k\nu_c\tau_p) \hat{f}_{bs}^{\star}(\nu + k\nu_c).
\tag{37}
$$

Here $\hat{x}(\nu)$ denotes the Fourier transform of a function $x(t)$, and we are using the fact that the Fourier transform of $\Pi(t;\tau)$ is given by $\tau \, \text{sinc}(\nu\tau)$, and the Fourier transform of $\text{III}(t;\tau)$ by $\tau^{-1} \text{III}(\nu;\tau^{-1})$, and $f_{bs}(\epsilon)$ and $p(\epsilon)$ are defined by (4) and (6), 325  respectively. The cardinal sine function is defined as

$$
\text{sinc}[(\cdot)] = \begin{cases} 1 & \text{if } \nu = 0 \\ \sin[\pi(\cdot)]/[\pi(\cdot)] & \text{if } \nu \neq 0 \end{cases},
\tag{38}
$$

and for the steps from (35) to (37) the convolution theorem is used. Expression (37) represents a series of amplitude modulated and shifted versions of the function $\hat{f}_{bs}^{\star}(\nu)$, which is obtained by taking the Fourier transform of (4),

$$
\hat{f}_{bs}^{\star}(\nu) = \text{sinc}(\nu\tau_s) \left[ 1 + i2\pi\nu\tau_b \right]^{-1} \exp(i2\pi\nu\tau_b),
\tag{39}
$$



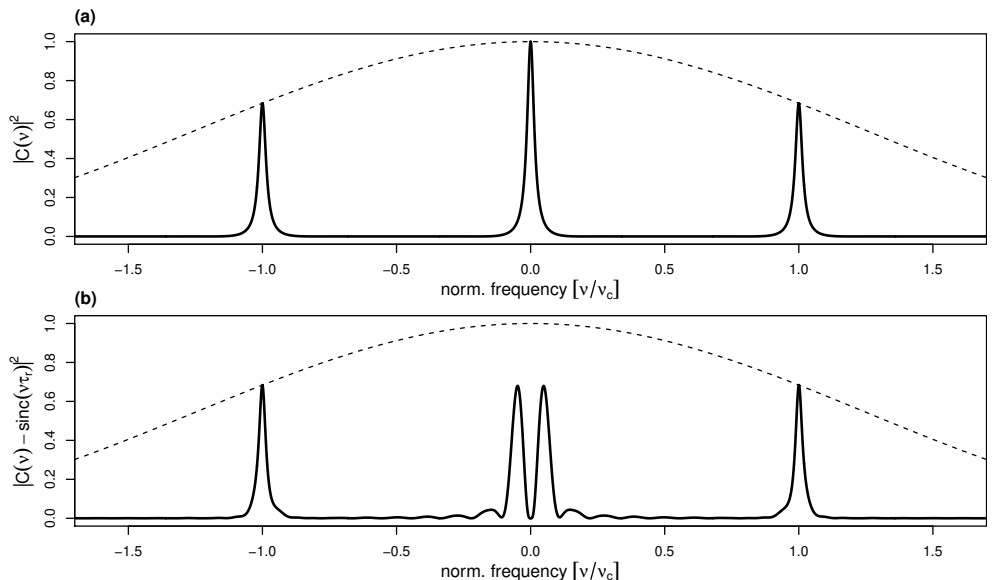

**Figure 2.** (a) Squared modulus of the characteristic function (or of the Fourier transform) of the jitter PDF, $|C(\nu)|^2$, given by (41) (solid line), for the same parameters as used in Fig. 1 (i.e., $\tau_b = 10$ yrs, $\tau_s = 5$ yrs, $\tau_p = 1/3$ yr); and the envelope function $\text{sinc}^2(\nu\tau_p)$ (dashed line). (b) Squared modulus of the error transfer function, $|C(\nu) - \text{sinc}(\nu\tau_r)|^2$, as it appears in (57), and with $\tau_r = 9$ yr; dashed line as in (a). The frequency axis is normalized by the seasonal cycle frequency $\nu_c = (1\text{ yr})^{-1}$.

and, thus,

$$|\hat{f}_{bs}^\star(\nu)|^2 = \text{sinc}^2(\nu\tau_s)\big[1 + (2\pi\nu\tau_b)^2\big]^{-1}, \qquad (40)$$

which is the product of a squared sinc-function and a Lorentzian function. In the following we assume that $[\max(\tau_b, \tau_s)]^{-1} \ll \nu_c$, such that the characteristic width of the functions $\hat{f}_{bs}^\star(\nu + k\nu_c)$ is much less than the shift increment $\nu_c$. Then these functions have negligible overlap and we can write

$$|C(\nu)|^2 = \sum_{k=-\infty}^{\infty} \text{sinc}^2(k\nu_c\tau_p)|\hat{f}_{bs}^\star(\nu + k\nu_c)|^2. \qquad (41)$$

The structure of $|C(\nu)|^2$ is illustrated by Fig. 2a, and it represents the squared modulus of the Fourier transform of the jitter PDF shown in Fig. 1a. Since it is shown for $\tau_b = 10$ yrs and $\tau_s = 5$ yrs, we have $[\max(\tau_b, \tau_s)]^{-1} = 1/(10\text{ yrs}) \ll \nu_c$ and, thus, the $|\hat{f}_{bs}^\star(\nu + k\nu_c)|^2$-peaks are well separated along the frequency axis. Finally, note, that the requirement, $\tau_s$ be a multiple of 1 year (made in section 2.3), implies that each of the peaks with $k \neq 0$ has one of its zeros at $\nu = 0$ because of the sinc-function

involved in (39). Thus, since $\hat{f}_{bs}^\star(\nu = 0) = 1$, it follows from (37) that $C(\nu = 0) = 1$ and, hence, from (35) with $\nu = 0$ that the jitter PDF $p(\epsilon)f_{bs}(\epsilon)$ does indeed integrate to unity.

To obtain the power spectral density of the reconstruction error components of $E_{X,n}$, we rewrite the integrand of (28) as $e^{i2\pi\nu t_n}e^{i2\pi\nu\epsilon_n^{(j)}}$ and split the jitter factor $e^{i2\pi\nu\epsilon_n^{(j)}}$ into its expected value, $C(\nu)$, and the deviation thereof, $e^{i2\pi\nu\epsilon_n^{(j)}} - C(\nu)$, as



in Moore and Thomson (1991). Then we can decompose $X_n^{(j)}$ as

$$X_n^{(j)} = U_n + V_n^{(j)}, \tag{42}$$

with the components

$$U_n = \int_{-\infty}^{\infty} e^{i2\pi\nu t_n} C(\nu) dZ(\nu) \tag{43}$$

and

$$V_n^{(j)} = \int_{-\infty}^{\infty} e^{i2\pi\nu t_n} \left[ e^{i2\pi\nu\epsilon_n^{(j)}} - C(\nu) \right] dZ(\nu). \tag{44}$$

From this we obtain, by analogy with the steps from (29) to (31), the auto-covariance functions of $U_n$ and $V_n^{(j)}$ as well as their cross-covariance function,

$$\langle U_n^\star U_{n'} \rangle_X = \int_{-\infty}^{\infty} e^{i2\pi\nu(t_{n'}-t_n)} |C(\nu)|^2 S_X(\nu) d\nu, \tag{45}$$

$$\langle\langle V_n^{(j)\star} V_{n'}^{(j)} \rangle_X \rangle_\epsilon = \int_{-\infty}^{\infty} e^{i2\pi\nu(t_{n'}-t_n)} \left[ C_{n,n'}(-\nu,\nu) - |C(\nu)|^2 \right] S_X(\nu) d\nu, \tag{46}$$

$$\langle\langle U_n^\star V_{n'}^{(j)} \rangle_X \rangle_\epsilon = \int_{-\infty}^{\infty} e^{i2\pi\nu(t_{n'}-t_n)} C^\star(\nu) \langle e^{i2\pi\nu\epsilon_n^{(j)}} - C(\nu) \rangle_\epsilon S_X(\nu) d\nu. \tag{47}$$

Since the term $\langle e^{i2\pi\nu\epsilon_n^{(j)}} - C(\nu) \rangle_\epsilon$ in (47) is zero, the cross power spectral density of $U_n$ and $V_n^{(j)}$ vanishes at all frequencies and, thus, the two processes are uncorrelated. Accordingly, the sum of their auto-covariance functions, (45) and (46), equals the auto-covariance function of $X_n^{(j)}$, given by (31). Furthermore, (31) with $n = n'$ shows that $\mathrm{Var}_{(X,\epsilon)}(X_n^{(j)}) = \mathrm{Var}_{(X)}(X)$.

Note, that the square bracket term in (46) represents the auto-covariance function of the jitter factor $e^{i2\pi\nu\epsilon_n^{(j)}}$, and (33) implies that

$$\left[ C_{n,n'}(-\nu,\nu) - |C(\nu)|^2 \right] = \begin{cases} 1 - |C(\nu)|^2 & \text{if } n = n' \\ 0 & \text{if } n \neq n' \end{cases}. \tag{48}$$

Thus, the auto-covariance function of $V_n^{(j)}$ is non-zero only at lag zero ($n = n'$) and zero at all other lags ($n \neq n'$) and, hence, $V_n^{(j)}$ is a white noise process. On the other hand, $U_n$ can be seen as the result of linearly filtering the signal $X(t)$ with the jitter PDF, $p(\epsilon)f_{bs}(\epsilon)$, and then subsampling it at the discrete times $t_n$, with $|C(\nu)|^2$ in (45) being interpreted as the squared modulus of the spectral transfer function. Since neither the linear filter nor the subsampling alters the expected value, we have $\langle U_n \rangle_X = \langle X_n^{(j)} \rangle_X = \langle X \rangle_X = 0$, and from (42) it follows that also $\langle V_n^{(j)} \rangle_X = \langle X_n^{(j)} - U_n \rangle_X = 0$. Finally, the stationarity





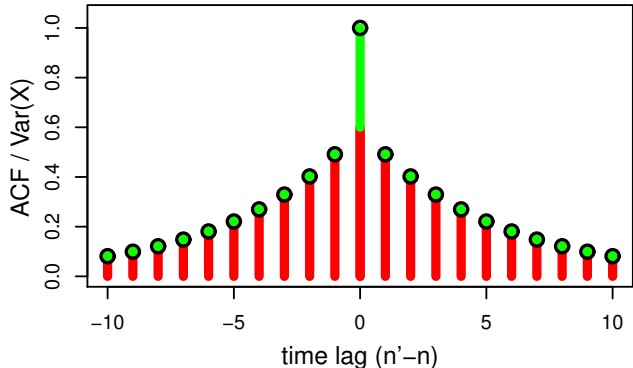

**Figure 3.** Schematic illustration of the auto-covariance function of the discrete process $X_n^{(j)}$ (black circles), as given by (31), normalized by the variance of $X(t)$; and the auto-covariance contribution from $U_n$ (red lines), as given by (45), and from $V_n^{(j)}$ (green lines), as given by (46), with green dots indicating zero contribution.

of $X$ and $\epsilon$ implies that $U_n$ and $V_n^{(j)}$ are stationary. The structure of the auto-covariance function of $X_n^{(j)}$, given by (31), is illustrated schematically by Fig. 3, highlighting its decomposition into the respective contributions from $U_n$ and $V_n^{(j)}$. In particular, it turns out that the magnitude of the variance of the white noise component $V_n^{(j)}$ is obtained by extrapolating the auto-covariance function from non-zero lags towards lag zero. This separates the full variance into two components (indicated

in the figure by the transition in color at lag zero), such that, from (45), (46) and (48),

$$\mathrm{Var}_{(X,\epsilon)}(V_n^{(j)}) = \mathrm{Var}_{(X)}(X) - \mathrm{Var}_{(X)}(U_n). \tag{49}$$

This is the key idea of the approach of Balakrishnan (1962), and we will return to this idea in section 3.2 in the context of the deterministic signal component $Y(t)$.

With these properties of the above components $U_n$ and $V_n^{(j)}$, we can now rewrite the error component $E_{X,n}$, defined by (14),

also using the $X$-component of (9), as

$$E_{X,n} = F_{X,n} + W_{X,n}, \tag{50}$$

with

$$F_{X,n} = U_n - \tilde{X}_n \tag{51}$$

and

$$W_{X,n} = \frac{1}{N} \sum_{j=1}^{N} V_n^{(j)}. \tag{52}$$

According to (11) and (12), a spectral representation of the $X$-component of the reference climate signal, $\tilde{X}_n$, is given by

$$\tilde{X}_n = \int_{-\infty}^{\infty} e^{i2\pi\nu t_n} \mathrm{sinc}(\nu\tau_r)dZ(\nu), \tag{53}$$





such that the auto-covariance function of $F_{X,n}$ is obtained from (43), (51) and (53) as

$$\left\langle F_{X,n}^{\star} F_{X,n'} \right\rangle_X = \int_{-\infty}^{\infty} e^{i2\pi\nu(t_{n'}-t_n)} |C(\nu) - \mathrm{sinc}(\nu\tau_r)|^2 S_X(\nu)d\nu. \tag{54}$$

Since, by analogy with (47), it can be shown that the cross power spectral density of $\tilde{X}_n$ and $V_n^{(j)}$ vanishes at all frequencies, the same holds for the error components $F_{X,n}$ and $W_{X,n}$ in (50) and, thus, the power spectral density of their sum equals the sum of their spectral densities. Finally, because $\epsilon_n^{(j)}$ is also white in terms of $j$, we have, also using (45) and (49),

$$\mathrm{Var}_{(X,\epsilon)}(W_{X,n}) = \left[ \mathrm{Var}_{(X)}(X) - \mathrm{Var}_{(X)}(U_n) \right]/N \tag{55}$$

$$= \frac{1}{N} \int_{-\infty}^{\infty} \left[ 1 - |C(\nu)|^2 \right] S_X(\nu)d\nu. \tag{56}$$

From this we can now obtain a spectral representation of the squared reconstruction uncertainty components $\mathcal{U}_{(1)}^2$ and $\mathcal{U}_{(2)}^2$, respectively, as defined by (24), by writing the power spectral density of $F_{X,n}$, denoted by $S_{\mathcal{U}_{(1)}}(\nu)$, and the power spectral density of $W_{X,n}$, denoted by $S_{\mathcal{U}_{(2)}}(\nu)$. Specifically, from (54) we obtain (also taking into account spectral aliasing and leakage, see Priestley, 1981, for example)

$$S_{\mathcal{U}_{(1)}}(\nu) = \mathrm{III}(\nu; \Delta t^{-1}) * \left\{ T\,\mathrm{sinc}^2(\nu T) * \left[ |C(\nu) - \mathrm{sinc}(\nu\tau_r)|^2 S_X(\nu) \right] \right\}, \tag{57}$$

with $-\nu_* < \nu \le \nu_*$. Here, $\nu_* = (2\Delta t)^{-1}$ denotes the Nyquist frequency, $\Delta t$ the sampling interval between the discrete sampling times $t_n = n\Delta t$, and $T$ the length of the proxy record (being a multiple of $\Delta t$). Likewise, by confining the variance of the white noise process $W_{X,n}$, given by (56), to the same frequency interval, we obtain the constant spectral density

$$S_{\mathcal{U}_{(2)}}(\nu) = \frac{\Delta t}{N} \int_{-\infty}^{\infty} \left[ 1 - |C(\nu')|^2 \right] S_X(\nu')d\nu', \tag{58}$$

with $-\nu_* < \nu \le \nu_*$. To understand the structure of $S_{\mathcal{U}_{(1)}}(\nu)$, note, that the term $|C(\nu) - \mathrm{sinc}(\nu\tau_r)|^2$ in (57), referred to as the
squared modulus of the error transfer function, acts as a linear filter on the stochastic component $X(t)$ of the supposed true climate signal. Its structure is illustrated by Fig. 2b, under the additional assumption that $\tau_r^{-1} \ll \nu_c$. It turns out that it represents a multi-bandpass filter, with the low-frequency band being confined between $[\max(\tau_b, \tau_s)]^{-1}$ and $\tau_r^{-1}$ (this corresponds to the frequency band of the transfer function discussed by Amrhein, 2019, see his Fig. 2), whereas each high-frequency band is confined to an interval bounded by $k\nu_c \pm [\max(\tau_b, \tau_s)]^{-1}$, with $k = \pm 1, \pm 2, \ldots$, according to (41). The consequences of this
particular filter structure, in conjunction with the effects of spectral aliasing, are discussed in section 4. Finally, according to the finite length of the proxy record, we need to subsample the above power spectral densities at the discrete frequencies $\nu_m = m\Delta\nu$ (with $m = 0, \pm 1, \pm 2, \ldots$ and $\Delta\nu = 1/T$), which yields

$$S_{\mathcal{U}_{(1)},m} = S_{\mathcal{U}_{(1)}}(\nu_m) \tag{59}$$





and

$$S_{\mathcal{U}_{(2)},m} = S_{\mathcal{U}_{(2)}}(\nu_m). \tag{60}$$

Since $F_{X,n}$ and $W_{X,n}$ have zero cross power spectral density, the power spectral density of $E_{X,n}$ is then given by

$$S_{\mathcal{U}_{(1,2)},m} = S_{\mathcal{U}_{(1)},m} + S_{\mathcal{U}_{(2)},m}. \tag{61}$$

## 3.2 Deterministic signal: Discrete orbital spectrum

The deterministic signal $Y(t)$, defined by (1), can be expressed as

$$Y(t) = Y_c(t)\big[1 + Y_a(t)\big], \tag{62}$$

with the seasonal cycle oscillation

$$Y_c(t) = \alpha_c\big[Y_c^-(t) + Y_c^+(t)\big] \tag{63}$$

and the amplitude modulating orbital oscillation

$$Y_a(t) = \alpha_a\big[Y_a^-(t) + Y_a^+(t)\big], \tag{64}$$

where

$$Y_c^{\pm}(t) = e^{\pm i(2\pi\nu_c t + \phi_c)}, \qquad Y_a^{\pm}(t) = e^{\pm i(2\pi\nu_a t + \phi_a)} \tag{65}$$

and

$$\alpha_c = \sigma_c/\sqrt{2}, \qquad \alpha_a = \sigma_a/\sqrt{2}. \tag{66}$$

Then we can rewrite the signal (62) as a complex Fourier series,

$$Y(t) = \mathcal{Y}^-(t) + \mathcal{Y}^+(t), \tag{67}$$

with

$$\mathcal{Y}^{\pm}(t) = \alpha_c\alpha_a Y_c^{\pm}(t)Y_a^{\mp}(t) + \alpha_c Y_c^{\pm}(t) + \alpha_c\alpha_a Y_c^{\pm}(t)Y_a^{\pm}(t), \tag{68}$$

such that $\mathcal{Y}^+(t) = [\mathcal{Y}^-(t)]^{\star}$. Thus, the right-hand side of (67) is the sum of six Fourier modes, two of which occur at the frequencies $\pm\nu_c$ (the carrier wave in amplitude modulation terminology) and four of which occur at the frequencies $\pm\nu_c \pm \nu_a$ (representing the sidebands of $\pm\nu_c$).

Again following the approach of Balakrishnan (1962), and by analogy with section 3.1, we evaluate $\big\langle Y_n^{(j)\star} Y_{n'}^{(j)} \big\rangle_\epsilon$, which, however, is not the auto-covariance function in this case, because sampling the seasonal cycle oscillation $Y_c(t)$ at an average





interval of $\Delta t$ may leave a non-zero bias, as $\Delta t$ is a multiple of 1 year $(= \nu_c^{-1})$. Thus, in general, we may have $\left\langle Y_n^{(j)} \right\rangle_\epsilon \neq 0$, and it turns out that

$$\left\langle Y_n^{(j)\star} Y_{n'}^{(j)} \right\rangle_\epsilon = \left\langle Y_n^{(j)} \right\rangle_\epsilon^\star \left\langle Y_{n'}^{(j)} \right\rangle_\epsilon + \mathrm{Cov}_{(\epsilon)}\left( Y_n^{(j)\star}, Y_{n'}^{(j)} \right), \tag{69}$$

where $\mathrm{Cov}_{(\epsilon)}(Y_n^{(j)\star}, Y_{n'}^{(j)})$ is the non-stationary auto-covariance function of $Y_n^{(j)}$. We now decompose this signal as

$$Y_n^{(j)} = A_n + B_n^{(j)}, \tag{70}$$

with the components

$$A_n = \left\langle Y_n^{(j)} \right\rangle_\epsilon \tag{71}$$

and

$$B_n^{(j)} = Y_n^{(j)} - \left\langle Y_n^{(j)} \right\rangle_\epsilon, \tag{72}$$

by analogy with the components $U_n$ and $V_n^{(j)}$, respectively, in section 3.1, such that we can rewrite (69) as

$$\left\langle Y_n^{(j)\star} Y_{n'}^{(j)} \right\rangle_\epsilon = A_n^\star A_{n'} + \mathrm{Cov}_{(\epsilon)}\left( B_n^{(j)\star}, B_{n'}^{(j)} \right). \tag{73}$$

The above also implies that $A_n$ and $B_n^{(j)}$ are uncorrelated.

The structure of $A_n$ is obtained from (67), by replacing $t$ in the exponential terms in (65) by $n\Delta t + \epsilon_n^{(j)}$ and then applying the expected value operator. Note, that because $\Delta t$ is a multiple of 1 year, the modes at $\pm\nu_c$ become aliases of $\nu = 0$, the modes at $\pm\nu_c - \nu_a$ become aliases of $\nu = -\nu_a$, and those at $\pm\nu_c + \nu_a$ become aliases of $\nu = \nu_a$. Then considering phase interference caused by the aliasing, using (34), (37) and (39), noting that $e^{\pm i2\pi\nu_c n\Delta t} = 1$, and exploiting the symmetry property $C(-k\nu_c + \nu) = C(k\nu_c + \nu)$, we obtain

$$A_n = 2\alpha_c \cos(\phi_c)\,\mathrm{sinc}(\nu_c\tau_p)\,A_n', \tag{74}$$

with

$$A_n' = 1 + \alpha_a \left( \hat{f}_{bs}^\star(-\nu_a) e^{-i(2\pi\nu_a t_n + \phi_a)} + \hat{f}_{bs}^\star(\nu_a) e^{i(2\pi\nu_a t_n + \phi_a)} \right). \tag{75}$$

If we explicitly express the argument and the modulus of $\hat{f}_b^\star(\nu_a)$ as

$$\phi_{b1} = \arg\left[ \hat{f}_b^\star(\nu_a) \right] = 2\pi\nu_a\tau_b - \arctan(2\pi\nu_a\tau_b) \tag{76}$$

and

$$M_{b1} = |\hat{f}_b^\star(\nu_a)| = \left[ 1 + (2\pi\nu_a\tau_b)^2 \right]^{-1/2}, \tag{77}$$



respectively, then we can rewrite (74), (75) as

$$A_n = \sigma_c \sqrt{2} \cos(\phi_c) \operatorname{sinc}(\nu_c \tau_p) A_n' \tag{78}$$

with

$$A_n' = 1 + \sigma_a \sqrt{2} M_{b1} \operatorname{sinc}(\nu_a \tau_s) \cos(2\pi \nu_a t_n + \phi_a + \phi_{b1}). \tag{79}$$

It turns out that, as long as we take the seasonal phase $\phi_c$ as fixed, $A_n$ represents a deterministic bias caused by uneven sampling of the seasonal cycle due to proxy seasonality, and that this bias varies in time because the amplitude of the seasonal cycle is modulated by orbital variations. Since $\sigma_a \sqrt{2} < 1$, the term $A_n'$ is always positiv and, thus, the sign of the bias $A_n$ is determined only by the seasonal phase $\phi_c$. Note, that the phase component $\phi_{b1}$ of the oscillation results from the asymmetry of the bioturbation PDF, $f_b(\epsilon)$, defined by (2), which creates a time lag caused by bioturbation. However, if $\tau_b \ll \nu_a^{-1}$, we have $\phi_{b1} \approx 0$ and the time lag vanishes.

To understand the structure of $B_n^{(j)}$ we consider its variance, given by the auto-covariance function at lag zero, $\operatorname{Cov}_{(\epsilon)}(B_n^{(j)\star}, B_{n'}^{(j)})|_{n=n'}$. From this we find, as is shown in the appendix, that $B_n^{(j)}$ is a non-stationary zero-mean white noise process. In particular, the variance has a stationary component, and two time-varying components oscillating at the frequencies $\nu_a$ and $2\nu_a$, respectively, see (A12). To illustrate this behaviour, we consider the following three simplified cases.

First, if there is no amplitude modulation of the seasonal cycle ($\sigma_a^2 = 0$), the variance of $B_n^{(j)}$ is stationary and is given by

$$\operatorname{Var}_{(\epsilon)}\left(B_n^{(j)}\right) = \sigma_c^2 \left\{ 1 - \operatorname{sinc}^2(\nu_c \tau_p) + \cos(2\phi_c)\left[\operatorname{sinc}(2\nu_c \tau_p) - \operatorname{sinc}^2(\nu_c \tau_p)\right] \right\}. \tag{80}$$

The dependence of this variance on the width of the proxy abundance window, $\tau_p$, and its seasonal timing, $\phi_c$, is illustrated by Fig. 4. If $\tau_p = 0$, the white noise variance vanishes because each year the same value is sampled from the seasonal cycle. If $\tau_p = 1$ yr, the white noise variance equals the seasonal cycle variance $\sigma_c^2$. For intermediate values of $\tau_p$ the white noise variance depends on the seasonal phase. Note, that for phases around $|\phi_c| = \pi/2$ the white noise variance can exceed the seasonal cycle variance by up to 22%.

Second, if the seasonal cycle is modulated by orbital variations ($0 < \sigma_a^2 < 1/2$), and we set $\tau_p = 0$ (and, for simplicity, we choose (i) $\tau_b = 0$ so as to avoid any additional phase lags, $\phi_{b1} = \phi_{b2} = 0$, and (ii) $\phi_c = 0$ for a maximum effect of proxy seasonality), then the variance of $B_n^{(j)}$ is given by

$$\operatorname{Var}_{(\epsilon)}\left(B_n^{(j)}\right) = 2\sigma_c^2 \sigma_a^2 \left\{ 1 - \operatorname{sinc}^2(\nu_a \tau_s) + \cos(4\pi \nu_a t_n + 2\phi_a)\left[\operatorname{sinc}(2\nu_a \tau_s) - \operatorname{sinc}^2(\nu_a \tau_s)\right] \right\}. \tag{81}$$

Note the analogy with (80), but with $\nu_c \tau_p$ and $\phi_c$ being replaced by $\nu_a \tau_s$ and $\phi_a$, respectively. This is because the white noise variance is now sampled from the orbital oscillation. Also it has now a time-varying component with frequency $2\nu_a$.

Third, if we consider the same case but with $\tau_p = 1$ yr, we obtain

$$\operatorname{Var}_{(\epsilon)}\left(B_n^{(j)}\right) = \sigma_c^2 \left\{ \left[1 + 2\sigma_a \sqrt{2} \operatorname{sinc}(\nu_a \tau_s) \cos(2\pi \nu_a t_n + \phi_a)\right] + \sigma_a^2 \left[1 + \operatorname{sinc}(2\nu_a \tau_s) \cos(4\pi \nu_a t_n + \phi_a)\right] \right\}. \tag{82}$$

In this case, the white noise variance has two time-varying components with frequencies $\nu_a$ and $2\nu_a$, respectively, because the amplitude modulation factor has the basic structure $1 + \cos(2\pi \nu_a t)$, the square of which, as it appears in the variance, is





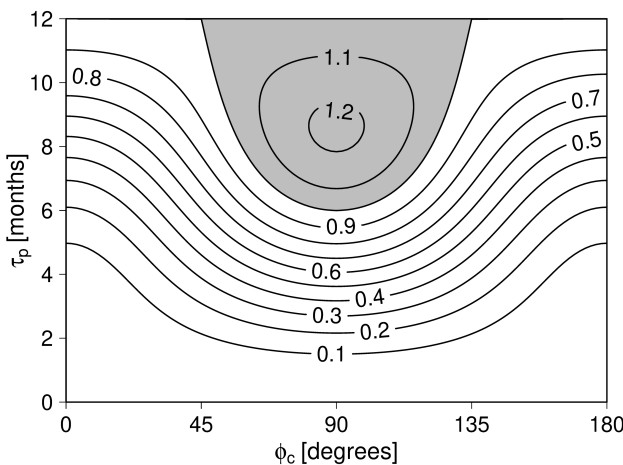

**Figure 4.** The white noise variance $\mathrm{Var}_{(\epsilon)}(B_n^{(j)})$ that is sampled from the seasonal cycle, normalized by the seasonal cycle variance $\sigma_c^2$, for the simplified case without amplitude modulation by orbital variations. The variance is shown as a function of the width of the proxy abundance window $\tau_p$ and of the seasonal phase $\phi_c$, which together characterize proxy seasonality. Values greater than one are indicated by gray shading.

$1 + 2\cos(2\pi\nu_a t) + \cos^2(2\pi\nu_a t)$. Note, that the seasonal phase $\phi_c$ has no effect in this case with $\tau_p$ covering the full seasonal cycle.

With these properties of the above components $A_n$ and $B_n^{(j)}$, we can now rewrite the error component $E_{Y,n}$, defined by (14), also using the $Y$-component of (9), as

$$E_{Y,n} = F_{Y,n} + W_{Y,n}, \tag{83}$$

with

$$F_{Y,n} = A_n, \tag{84}$$

because $\tilde{Y}_n = 0$, and

$$W_{Y,n} = \frac{1}{N}\sum_{j=1}^{N} B_n^{(j)}. \tag{85}$$

Then we obtain the reconstruction bias, defined by (22), as

$$\mathcal{B}_n = \langle A_n \rangle_{\phi_c} = \sigma_c \sqrt{2}\,\gamma_1 A_n', \tag{86}$$

with

$\gamma_1 = \cos(\langle\phi_c\rangle_{\phi_c})\,\mathrm{sinc}(\Delta_{\phi_c}/2\pi)\,\mathrm{sinc}(\nu_c\tau_p),$ \hfill (87)





where the seasonal phase uncertainty $\Delta_{\phi_c}$ and the expected seasonal cycle phase $\langle\phi_c\rangle_{\phi_c}$ are defined by (18), and $A'_n$ by (79). With this, the third squared reconstruction uncertainty component, defined by (24), can be expressed as

$$\mathcal{U}^2_{(3),n} = \left\langle A_n^2 \right\rangle_{\phi_c} - \left\langle A_n \right\rangle^2_{\phi_c} = \sigma_c^2 \gamma_2 A'^2_n. \tag{88}$$

with

$$\gamma_2 = \left\{ 1 - \mathrm{sinc}^2(\Delta_{\phi_c}/2\pi) + \cos(2\langle\phi_c\rangle_{\phi_c})\left[ \mathrm{sinc}(2\Delta_{\phi_c}/2\pi) - \mathrm{sinc}^2(\Delta_{\phi_c}/2\pi) \right] \right\} \mathrm{sinc}^2(\nu_c \tau_p). \tag{89}$$

Finally, from (85), and because $\epsilon_n^{(j)}$ is white in terms of $j$, we have

$$\mathrm{Var}_{(\epsilon)}(W_{Y,n}) = \mathrm{Var}_{(\epsilon)}(B_n^{(j)})/N \tag{90}$$

and, thus, the fourth squared reconstruction uncertainty component is obtained as

$$\mathcal{U}^2_{(4),n} = \frac{1}{N}\left\langle \mathrm{Var}_{(\epsilon)}\left( B_n^{(j)} \right) \right\rangle_{\phi_c}, \tag{91}$$

with $\mathrm{Var}_{(\epsilon)}(B_n^{(j)})$ given by (A12), and where applying the expected value operator $\langle\cdot\rangle_{\phi_c}$ amounts to replacing each instance of $\cos(2\phi_c)$, as it appears multiple times in the components of (A12), according to

$$\cos(2\phi_c) \xrightarrow{\text{repl.}} \cos(2\langle\phi_c\rangle_{\phi_c})\,\mathrm{sinc}(2\Delta_{\phi_c}/2\pi). \tag{92}$$

Note, that $\mathrm{sinc}(2\Delta_{\phi_c}/2\pi) = 0$ if $\Delta_{\phi_c} = \pi$ or $2\pi$, in which case the expected value in (91) simplifies, because all terms in the components of (A12) that are multiplied by $\cos(2\phi_c)$ vanish.

To obtain spectral representations of $\mathcal{B}_n^2$ and $\mathcal{U}^2_{(3),n}$, we consider first the power spectral density of the signal $A'_n$, limited to a finite time interval of length $T$ (centered at $t = 0$), interpreted as the length of the proxy record. Specifically, we can express this discretized power spectral density as

$$S_{0,m} = \left| \mathcal{F}\left[ \Pi(t_n; T/\Delta t)A'_n \right](\nu_m) \right|^2 / T, \tag{93}$$

given at the discrete frequencies $\nu_m = m\Delta\nu$ (with $m = 0, \pm 1, \pm 2, \ldots$ and $\Delta\nu = 1/T$). In the above,

$$\mathcal{F}[x_n](\nu) = \sum_{n=-n_h}^{n_h} e^{-i2\pi\nu n\Delta t} x_n \Delta t \tag{}$$

denotes the discrete time Fourier transform of a sequence $x_n$ (with $-\nu_* < \nu \le \nu_*$), and the discrete rectangle function acts as a window to confine $A'_n$ to the finite time interval, and $T/\Delta t$ is the number of sampling times $t_n = n\Delta t$, with $n = 0, \pm 1, \pm 2, \ldots, \pm n_h$, where $n_h = (T/\Delta t - 1)/2$, for odd numbers of sampling times.

The discrete time Fourier transform of the rectangle function is given by the Dirichlet kernel (see Priestley, 1981, p. 437) which can be expressed as $T$-times the aliased (or periodic) sinc-function, defined by

$$\mathrm{asinc}(\nu; T, \Delta t) = \begin{cases} 1 & \text{if } \nu = 0 \\ \sin(\pi\nu T)/[\sin(\pi\nu\Delta t)T/\Delta t] & \text{if } \nu \ne 0 \end{cases}, \tag{94}$$




within the interval $-2\nu_* < \nu < 2\nu_*$. With this, and if we express the discrete time Fourier transform of $A'_n$ as a series of Dirac delta functions, we obtain from (75),

$$\mathcal{F}\big[\Pi(t_n;T/\Delta t)A'_n\big](\nu) = T\,\mathrm{asinc}(\nu;T,\Delta t) * \big\{\delta(\nu) + \alpha_a\big[\delta(\nu+\nu_a)\hat{f}^\star_{bs}(-\nu_a)e^{-i\phi_a} + \delta(\nu-\nu_a)\hat{f}^\star_{bs}(\nu_a)e^{i\phi_a}\big]\big\}. \tag{95}$$

Then from (93), also considering phase interference between the asinc-functions, using (76) and (77), and noting that the central asinc-function centered at $\nu = 0$ has its zeros at the discrete frequencies $\nu_m$, we have

$$S_{0,m} = T(S_{c,m} + S_{ca,m} + S_{a,m}), \tag{96}$$

with

$$S_{c,m} = \delta_m, \qquad S_{ca,m} = \delta_m \sigma_a \sqrt{2} M_{b1}\,\mathrm{sinc}(\nu_a\tau_s)2\xi(0)\cos(\phi_a+\phi_{b1}), \tag{97}$$

and

$$S_{a,m} = (\sigma_a^2/2)M_{b1}^2\,\mathrm{sinc}^2(\nu_a\tau_s)\big\{\xi_+^2(\nu_m)+\xi_-^2(\nu_m)+2\xi_+(\nu_m)\xi_-(\nu_m)\cos\big[2(\phi_a+\phi_{b1})\big]\big\}, \tag{98}$$

and where

$$\xi_\pm(\nu) = \mathrm{asinc}(\nu\pm\nu_a;T,\Delta t), \tag{99}$$

and $\delta_m$ denotes the single-argument Kronecker delta, with $\delta_{m=0} = 1$ and $\delta_{m\neq0} = 0$.

With (96) we obtain, by analogy with (86) and (88), the spectral representation of $\mathcal{B}_n^2$, given by

$$S_{\mathcal{B},m} = 2\sigma_c^2\gamma_1^2 S_{0,m}, \tag{100}$$

with $-\nu_* < \nu_m \leq \nu_*$, and the spectral representation of $\mathcal{U}_{(3),n}^2$, given by

$$S_{\mathcal{U}_{(3)},m} = \sigma_c^2\gamma_2 S_{0,m}. \tag{101}$$

with $-\nu_* < \nu_m \leq \nu_*$. Since $W_{Y,n}$, given by (85), is white noise, we can express the spectral representation of $\mathcal{U}_{(4),n}^2$ as

$$S_{\mathcal{U}_{(4)},m} = \frac{\Delta t}{N}\Big\langle\overline{\mathrm{Var}}_{(\epsilon)}\big(B_n^{(j)}\big)\Big\rangle_{\phi_c}, \tag{102}$$

with $-\nu_* < \nu_m \leq \nu_*$, and where $\overline{\mathrm{Var}}_{(\epsilon)}(B_n^{(j)})$ is given by (A19). Note, that, as in (91), applying the expected value operator $\langle\cdot\rangle_{\phi_c}$ amounts to applying the replacement (92) to the components of (A19). From the above, the power spectral density of $E_{Y,n}$ is then given by

$$S_{\mathcal{B},\mathcal{U}_{(3,4)},m} = S_{\mathcal{B},m} + S_{\mathcal{U}_{(3)},m} + S_{\mathcal{U}_{(4)},m}. \tag{103}$$

Hence, we obtain the spectral representation of the squared error equation (21) that relates the RMS reconstruction error $\mathcal{E}_n$, defined by (19), to the reconstruction bias $\mathcal{B}_n$ and the reconstruction uncertainty $\mathcal{U}_n$,

$$S_{\mathcal{E},m} = S_{\mathcal{B},m} + S_{\mathcal{U},m}, \tag{104}$$

where $S_{\mathcal{U},m} = S_{\mathcal{U}_{(1)},m} + S_{\mathcal{U}_{(2)},m} + S_{\mathcal{U}_{(3)},m} + S_{\mathcal{U}_{(4)},m}$, the first two components of which are given by (59) and (60), respectively, at the end of section 3.1. Thus, $S_{\mathcal{E},m}$ is the power spectral density (given at the discrete frequencies $\nu_m$) of the reconstruction
error $E_n$ (given at the discrete times $t_n$).





## 4 Timescale-dependent reconstruction uncertainty

The reconstruction uncertainty components $\mathcal{U}_{(1)}$, $\mathcal{U}_{(2)}$, $\mathcal{U}_{(3),n}$, $\mathcal{U}_{(4),n}$ and the reconstruction bias $\mathcal{B}_n$, defined in section 2.5, can now be quantified using the expressions derived in section 3. Specifically, given the set of parameters of the reconstruction uncertainty model (see Table 1), including the specifications of the deterministic component of the supposed true climate signal,

and given the power spectral density of the stochastic signal component, we obtain

- the uncertainty component $\mathcal{U}_{(1)}$ that arises from the various smoothing processes affecting the stochastic signal component $X(t)$, in the limit of infinitely many signal carriers retrieved from each slice of sediment ($N \to \infty$). From (59), or from (57), noting that for stochastic signals spectral aliasing and leakage do neither generate nor destroy, but only redistribute power spectral density, we have

$$\mathcal{U}_{(1)}^2 = \sum_{m=-m_h}^{m_h} S_{\mathcal{U}_{(1)},m}\Delta\nu \tag{105}$$

$$= \int_{-\infty}^{\infty} |C(\nu) - \operatorname{sinc}(\nu\tau_r)|^2 S_X(\nu)d\nu, \tag{106}$$

with $m_h = (T/\Delta t - 1)/2$, for odd numbers of sampling times, and $C(\nu)$ is given by (37). This uncertainty component depends on the timescale parameters $\tau_b$, $\tau_s$, $\tau_r$, $\tau_p$, and on $S_X(\nu)$.

- the white noise uncertainty component $\mathcal{U}_{(2)}$ that arises from sampling only a finite number $N$ of signal carriers from
each slice of sediment. From (60), or likewise from (58), we obtain

$$\mathcal{U}_{(2)}^2 = \sum_{m=-m_h}^{m_h} S_{\mathcal{U}_{(2)},m}\Delta\nu \tag{107}$$

$$= \frac{1}{N} \int_{-\infty}^{\infty} \left[1 - |C(\nu)|^2\right] S_X(\nu)d\nu. \tag{108}$$

This uncertainty component depends on the timescale parameters $\tau_b$, $\tau_s$, $\tau_p$, and on $S_X(\nu)$ and $N$.

- the reconstruction bias $\mathcal{B}_n$, its uncertainty $\mathcal{U}_{(3),n}$, caused by the imperfectly known seasonal timing of the proxy abun-
dance, and the white noise uncertainty component $\mathcal{U}_{(4),n}$ that arises from sampling only a finite number of signal carriers, which are readily given in the time domain by (86), (88) and (91), respectively. These components depend on the timescale parameters $\tau_b$, $\tau_s$, $\tau_p$, on the seasonal phase parameters $\langle\phi_c\rangle_{\phi_c}$, $\Delta_{\phi_c}$, and on the specifications of the deterministic signal component, $\sigma_c$, $\sigma_a$, $\nu_c$, $\nu_a$ and $\phi_a$. The white noise component $\mathcal{U}_{(4),n}$ also depends on $N$. The time averages of the squares of these components, over the length of the proxy record $T$, can be obtained directly from their spectral
representations, given by (100), (101) and (102), respectively, as

$$\overline{\mathcal{B}_n^2} = \sum_{m=-m_h}^{m_h} S_{\mathcal{B},m}\Delta\nu, \qquad \overline{\mathcal{U}_{(3),n}^2} = \sum_{m=-m_h}^{m_h} S_{\mathcal{U}_{(3)},m}\Delta\nu, \qquad \overline{\mathcal{U}_{(4),n}^2} = \sum_{m=-m_h}^{m_h} S_{\mathcal{U}_{(4)},m}\Delta\nu, \tag{109}$$





which then also depend on $T$.

Since all of the above uncertainty components as well as the bias depend on a number of timescale parameters, the RMS reconstruction error $\mathcal{E}_n$, in this sense, already represents a timescale-dependent uncertainty measure. However, we may extend the concept of uncertainty timescale dependence as follows.

In practice, during the process of data analysis, climate reconstructions are often smoothed by some linear filter, either because one is explicitly interested in time averages of the reconstructed climate variable, or because one may hope to reduce the reconstruction uncertainty by averaging out short-timescale noise. However, the extent to which the uncertainty actually shrinks depends on the auto-correlation structure of the reconstruction error, which, by the Wiener-Khintchine theorem, is directly related to the power spectral density of the error. Thus, from the expressions of the error power spectral densities, derived in section 3, we can directly quantify the uncertainty reduction that is achieved by applying a linear filter, as is shown in the following for the uncertainty components $\mathcal{U}_{(1)}$ and $\mathcal{U}_{(2)}$, as an example.

If the reconstruction error time series is smoothed, for simplicity, by a discrete moving average filter of width $\tau_0$ (being a multiple of $\Delta t$), then the squared uncertainty, obtained after smoothing, is given by

$$\mathcal{U}_{(1,2)}^2(\tau_0) = \sum_{m=-m_h}^{m_h} \mathrm{asinc}^2(\nu_m; \tau_0, \Delta t) S_{\mathcal{U}_{(1,2)},m} \Delta \nu, \tag{110}$$

where the squared $\mathrm{asinc}$-function represents the squared modulus of the discrete time Fourier transform of the filter window, acting as a spectral transfer function, and the $\mathrm{asinc}$-function is defined by (94). Note, that if $\tau_0 = \Delta t$ (i.e., no smoothing), then this transfer function is constantly one across all frequencies, and if $\tau_0 = T$, then it is equal to one at frequency zero, and zero at all other frequencies. Thus, in the latter case the uncertainty of the time average over the entire proxy record is obtained. Fig. 5 illustrates the above for some choice of parameters, designed to exmplify as many aspects as possible of the uncertainty estimation procedure in a single example, at the expense of using somewhat unrealistic parameter values. More realistic application examples of the method follow in Part II of this study (Dolman et al., 2019). Specifically, we set $\tau_b = 10$ yrs, $\tau_s = \tau_r = 6$ yrs, $\tau_p = (1/3)$ yr, $\Delta t = 6$ yrs, $T = 23\Delta t = 138$ yrs, $\tau_0 = 3\Delta t = 18$ yrs, and $N = 100$. For the power spectral density of $X(t)$ we assume a Lorentzian shaped AR(1) red noise spectrum, given by $S_X(\nu) = 2\alpha/[(2\pi\nu)^2 + \alpha^2]$, with the characteristic timescale $\alpha^{-1} = (1/10)$ yr, such that the process $X$ is only weakly red.

This power spectral density is shown in Fig. 5a by the gray line. According to the reconstruction uncertainty model, defined in section 2, $S_X(\nu)$ is decomposed into two components: (i) $|C(\nu)|^2 S_X(\nu)$, shown by the red line, the integral of which equals the variance of $U_n$, defined by (43), where $|C(\nu)|^2$ (shown in Fig. 2a) acts as a spectral transfer function on $S_X(\nu)$; (ii) $[1 - |C(\nu)|^2] S_X(\nu)$, the integral of which, indicated by the green area, equals the variance of the white noise component $V_n^{(j)}$, defined by (44). If $S_X(\nu)$ is multiplied by the squared modulus of the error transfer function (shown in Fig. 2b), the component $|C(\nu) - \mathrm{sinc}(\nu\tau_r)|^2 S_X(\nu)$, shown by the blue line, is obtained, the integral of which equals the variance of $F_{X,n}$, defined by (51).

This component (blue dots) as well as the white noise component (green dots) are shown again in Fig. 5b, but after spectral aliasing and leakage have been applied, according to the sampling and measurement procedure described in section 2.3. These





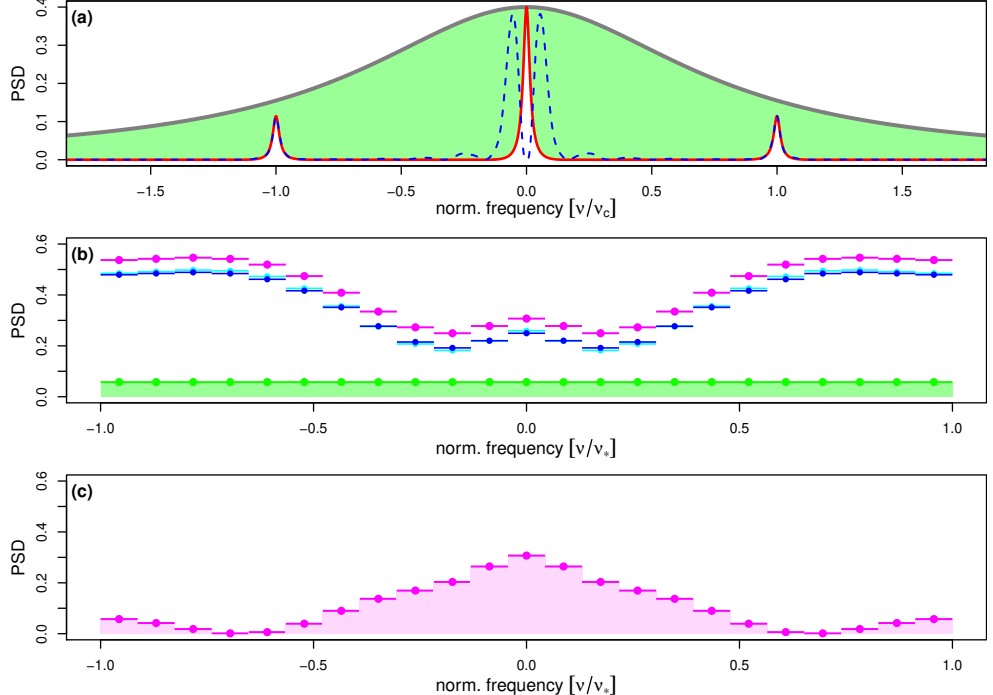

**Figure 5.** Illustration of the method for estimating timescale-dependent reconstruction uncertainties in the spectral domain; shown for the uncertainty components $\mathcal{U}_{(1)}$ and $\mathcal{U}_{(2)}$ which are based on the stochastic component $X(t)$ of the supposed true climate signal. (a) Power spectral density $S_X(\nu)$ (gray line) of this signal component, defined on a continuous and infinite frequency axis, normalized by the seasonal cycle frequency $\nu_c = (1 \text{ yr})^{-1}$; together with the product $|C(\nu)|^2 S_X(\nu)$ (red line); and the product $|C(\nu) - \text{sinc}(\nu\tau_r)|^2 S_X(\nu)$ (blue line). The green area equals the integral $\int_{-\infty}^{\infty}[1 - |C(\nu)|^2]S_X(\nu)d\nu$, as it appears in (58), which measures the variance of the white noise error component caused by sampling only a finite number of signal carriers. (b) The same white noise variance (green area), but divided by $N$ (the number of signal carriers) and after spectral aliasing and leakage have been applied, to obtain the power spectral density $S_{\mathcal{U}_{(2)},m}$ (green dots), defined by (60), on a finite and discrete frequency axis, now normalized by the Nyquist frequency $\nu_* = (2\Delta t)^{-1}$; together with $S_{\mathcal{U}_{(1)},m}$ (blue dots), defined by (59); and $S_{\mathcal{U}_{(1,2)},m}$ (magenta dots), defined by (61). Cyan dots indicate the same as blue dots, but neglecting the effect of spectral leakage for comparison. (c) The product $\text{asinc}^2(\nu_m; \tau_0, \Delta t)S_{\mathcal{U}_{(1,2)},m}$ (magenta dots), the integral of which (magenta area) equals the squared reconstruction uncertainty $\mathcal{U}_{(1,2)}^2(\tau_0)$, defined by (110).

components represent the discretized power spectral densities $S_{\mathcal{U}_{(1)},m}$ and $S_{\mathcal{U}_{(2)},m}$, respectively. Note, that the broad peaks at non-zero frequencies in Fig. 5b are direct images of the low-frequency peaks in Fig. 5a (blue line), whereas the bump centered at $\nu = 0$ represents the summed aliases of the high-frequency peaks in Fig. 5a (blue line) at $\pm k\nu_c$. Without proxy seasonality ($\tau_p = 1$ yr) those peaks do not exist and, thus, the $S_{\mathcal{U}_{(1)},m}$ power spectrum in Fig. 5b falls off to near zero at $\nu = 0$. Only spectral leakage may then lead to non-zero $S_{\mathcal{U}_{(1)},m}$ at $\nu = 0$, although in the example shown here the effect of the leakage is

small (cyan dots). However, in cases with small $T$ (implying large $\Delta\nu$) spectral leakage can provide a relevant contribution of power at $\nu = 0$, as the power from the neighboring broad spectral peaks is then effectively redistributed to the center of the





frequency domain. This contribution is particularly important if the uncertainty of the time average over the entire proxy record is computed, by setting $\tau_0 = T$ in (110), as in this case it is the only contribution to $\mathcal{U}_{(1)}^2(\tau_0)$.

Finally, the sum of the above components, given by $S_{\mathcal{U}_{(1,2)},m}$, is shown by the magenta dots in Fig. 5c. Multiplying this

summed power spectral density by the spectral transfer function of the discrete moving average window, mentioned above, and then integrating, yields the squared reconstruction uncertainty after smoothing, $\mathcal{U}_{(1,2)}^2(\tau_0)$, defined by (110), and shown by the magenta area in Fig. 5c.

Likewise, one can define other timescale-dependent uncertainty metrics. For example, one might be interested in the uncertainty of the difference between the time averages over two periods of length $T_1$ and $T_2$, which are separated in time by the

interval $\delta t = (n' - n)\Delta t$, measured from center to center. This can be expressed by the variance

$$\text{Var}\left(\left[T_1^{-1}\Pi(t_n; T_1/\Delta t) * E_{X,n}\right] - \left[T_2^{-1}\Pi(t_n; T_2/\Delta t) * E_{X,n+n'}\right]\right) =$$

$$\text{Var}\left(T_1^{-1}\Pi(t_n; T_1/\Delta t) * E_{X,n}\right) + \text{Var}\left(T_2^{-1}\Pi(t_n; T_2/\Delta t) * E_{X,n+n'}\right)$$

$$-2\,\text{Cov}\left(T_1^{-1}\Pi(t_n; T_1/\Delta t) * E_{X,n}, T_2^{-1}\Pi(t_n; T_2/\Delta t) * E_{X,n+n'}\right), \tag{111}$$

if it were to be computed for the uncertainty components based on $X(t)$, and where $T_1$, $T_2$ and $\delta t$ are multiples of $\Delta t$. Then, using the Wiener-Khintchine theorem, we obtain the difference uncertainty metric

$$\delta\mathcal{U}_{(1,2)}^2(T_1, T_2, \delta t) = \sum_{m=-m_h}^{m_h}\left[\text{asinc}^2(\nu_m; T_1, \Delta t) + \text{asinc}^2(\nu_m; T_2, \Delta t)\right]S_{\mathcal{U}_{(1,2)},m}\Delta\nu$$

$$-2\mathcal{F}^{-1}\left[\text{asinc}(\nu_m; T_1, \Delta t)\,\text{asinc}(\nu_m; T_2, \Delta t)S_{\mathcal{U}_{(1,2)},m}\right](\delta t), \tag{112}$$

where $\mathcal{F}^{-1}[x_m](k\Delta t) = \sum_{m=-m_h}^{m_h} e^{i2\pi m\Delta\nu k\Delta t}x_m\Delta\nu$ denotes the inverse discrete Fourier transform of a sequence $x_m$. Note, that in this form the above difference uncertainty metric is valid only for stationary uncertainty components. If the seasonal cycle amplitude is constant over time (i.e., no orbital variations), then all uncertainty components are stationary. If orbital variations are taken into account, however, only $\mathcal{U}_{(1)}$ and $\mathcal{U}_{(2)}$ are stationary, as shown in section 3. For the components $\mathcal{U}_{(3),n}$, $\mathcal{U}_{(4),n}$ and $\mathcal{B}_n$ the difference metric, in this case, had to be computed directly from their time domain expressions, see (86),

(88) and (91), respectively.

To conclude this section, we briefly present an example of the time series and power spectra of the reconstruction bias $\mathcal{B}_n$ and of the reconstruction uncertainty components $\mathcal{U}_{(3),n}$ and $\mathcal{U}_{(4),n}$. Specifically, we set $\tau_b = \tau_s = 100$ yrs, $\tau_p = (1/3)$ yr, $\Delta t = 100$ yrs, $T = 101\Delta t = 10100$ yrs, and $N = 5$. Furthermore, the deterministic signal is specified, in this example, by the parameters $\sigma_c = \sqrt{1/2}$, $\langle\phi_c\rangle_{\phi_c} = \pi/4$, $\Delta_{\phi_c} = \pi/2$, $\sigma_a = \sqrt{1/8}$, $\nu_a = (23\,\text{kyrs})^{-1}$, and $\phi_a = \pi/2$, which implies that the

amplitude of the seasonal cycle decreases during the 10100 years, as it is the case during the Holocene. According to the proxy seasonality parameter values chosen here, the bias $\mathcal{B}_n$ is positive and exhibits a negative trend, as shown in Fig. 6a (cyan line). Likewise the uncertainty components $\mathcal{U}_{(3),n}$ and $\mathcal{U}_{(4),n}$ also decrease over time (blue and green lines). Since the orbital modulation frequency $\nu_a$ is located on the discrete frequency axis between $\nu_0$ and $\nu_1$, its power spectral density is distributed by spectral leakage across all frequencies. This yields a highly red power spectrum of $\mathcal{B}_n$ and $\mathcal{U}_{(3),n}$, shown in



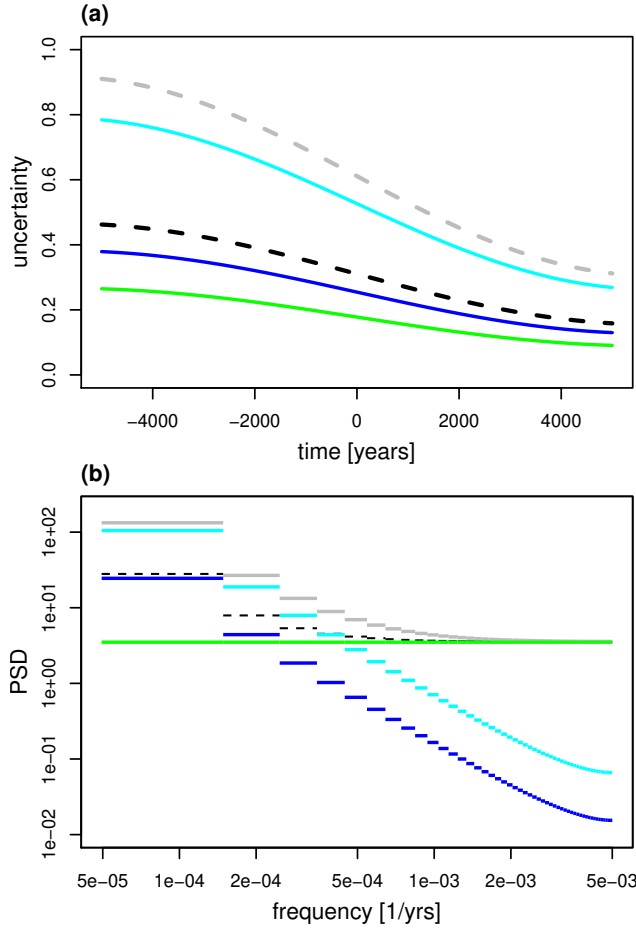

**Figure 6.** Example of the reconstruction uncertainty based on the deterministic component $Y(t)$ of the supposed true climate signal. (a) Time series of the reconstruction bias $\mathcal{B}_n$ (cyan line), of the uncertainty components $\mathcal{U}_{(3),n}$ (blue line) and $\mathcal{U}_{(4),n}$ (green line), of $[\mathcal{U}^2_{(3,4),n}]^{1/2}$ (dashed black line), and $[\mathcal{B}^2_n + \mathcal{U}^2_{(3,4),n}]^{1/2}$ (dashed gray line). (b) Power spectral densities of the corresponding error components, $S_{\mathcal{B},m}$ (cyan line), $S_{\mathcal{U}_{(3)},m}$ (blue line), $S_{\mathcal{U}_{(4)},m}$ (green line), $S_{\mathcal{U}_{(3,4)},m}$ (dashed black line), and $S_{\mathcal{B},\mathcal{U}_{(3,4)},m}$ (gray line)

Fig. 6b. Thus, at high frequencies the summed power spectral densities $S_{\mathcal{U}_{(3,4)},m}$ (dashed black line) and $S_{\mathcal{B},\mathcal{U}_{(3,4)},m}$ (dashed gray line) are dominated by the white noise component, whereas at low frequencies they are dominated by the effect of orbital variations. Hence, if we were to compute the timescale-dependent uncertainty metric (110), but for $\mathcal{U}_{(3),n}$ and $\mathcal{U}_{(4),n}$, denoted by $\mathcal{U}^2_{(3,4),n}(\tau_0)$, the uncertainty would shrink only slowly for increasing values of $\tau_0$, because the orbital variations are associated with a highly correlated error in time at long timescales.





## 5 Discussion

To allow for an analytic treatment of the problem, the method for estimating timescale-dependent reconstruction uncertainties, presented in sections 2 to 4, is necessarily based on a number of simplifying assumptions. Specifically,

- we assume a fixed proxy seasonality in the sense of applying every year the same seasonal timing of a prescribed proxy abundance period, characterized by the parameters $\tau_p$ and $\phi_c$. For this reason we have to separate the supposed true climate signal into a stochastic component $X(t)$, and a deterministic component $Y(t)$ that represents the seasonal cycle, because proxy seasonality then implies an in-phase subsampling from $Y(t)$ which, in turn, affects the amount of variance aliased from the seasonal cycle, $\mathcal{U}_{(4),n}$, and which may also lead to a reconstruction bias $\mathcal{B}_n$ and associated uncertainty $\mathcal{U}_{(3),n}$. This scenario represents the extreme case where a seasonal abundance period is completely imposed on the proxy by an external process (see, for example, Leduc et al., 2010), such as, for example, seasonally determined nutrient supply, possibly controlled by the seasonality of solar irradiance or oceanic upwelling. By contrast, in the opposite extreme case where no seasonality is imposed at all, we do not need to separate the climate signal into $X(t)$ and $Y(t)$. In this case the total climate signal is fully recorded by the proxy, but its total variance is reduced by some factor because of habitat tracking if the habitat PDF of the proxy is narrower than the PDF of the climate signal. According to the idea of Mix (1987), this factor can be obtained by multiplying the two PDFs, and it may possibly also be expressed as a frequency dependent spectral transfer function. This scenario corresponds to setting $\tau_p = 1$ yr in our method, and subsequently multiplying the obtained error power spectrum by the aforementioned transfer function. Hence, if we introduce some parameter, $0 \leq s \leq 1$, that measures the extent to which seasonality is imposed for a specific proxy record (with $s = 0$ indicating no imposition of seasonality), then we may express the actual uncertainty as a linear combination of the uncertainties obtained from the above two extreme scenarios, weighted by $s$ and by $1 - s$, respectively. Note, however, that the effects of seasonality can be rather complex (see, for example, Jonkers and Kučera, 2017), depending on the type of proxy used, and, thus, the optimal strategy for modelling the associated uncertainties depends on the specific application.

- we neglect calibration errors, representing uncertainties regarding the climate-proxy relationship. Assuming this relationship is linear and is obtained by linear regression, errors of this type may have two effects. Uncertainties in the intercept parameter will introduce a reconstruction uncertainty that is constant in time like the bias uncertainty $\mathcal{U}_{(3),n}$ (in the case without orbital variations). Uncertainties in the slope parameter, on the other hand, will introduce a frequency independent uncertainty in the error variance. The mean of the possible error variances, however, might be close to the variance obtained from our method, unless the PDF of the obtained error variances is strongly skewed. If the climate-proxy relationship is non-linear, or if there are uncertainties regarding the linearity itself, modelling of the implied uncertainties might be more complex, although it should still be possible to decompose those errors into a bias and a variance component.





    – we assume a constant sediment accumulation rate and a constant bioturbation depth, and we also assume regular sampling from the sediment core, and we neglect dating uncertainties, although relaxing these assumtions may generate additional uncertainties of noticable magnitude. For example, the relevance of dating uncertainties is demonstrated by Goswami et al. (2014) and Boers et al. (2017). If these sources of uncertainty are treated in a stochastic sense, they could, in principle, be included into our approach by allowing for correlated sampling jitter $\epsilon$, the mathematical basis of which is given by Balakrishnan (1962), see also Moore and Thomson (1991). More generally, these uncertainties could be modelled by allowing for a variable depth-time relationship, and perhaps by also allowing for non-stationarity of the uncertainty components $\mathcal{U}_1$ and $\mathcal{U}_2$ to represent variations of the smoothing timescales $\tau_b$ and $\tau_s$.

From the above it turns out that, in its current form, the method is neither complete, in terms of processes affecting the reconstruction uncertainty, nor does it cover all possible reconstruction scenarios, in terms of proxy type and application context. However, our formulation of the method outlines a conceptually and mathematically well-founded approach of how timescale-dependent reconstruction uncertainties could, and probably should, be estimated—in particular, when systematic and exact quantification is required. This latter point is highly relevant, for example, in the context of comparisons between circulation models and paleo-observations (e.g., Lohmann et al., 2013; Laepple and Huybers, 2014; Matsikaris et al., 2016), or likewise for any reanalysis efforts (e.g., Hakim et al., 2016), if data obtained from proxy records are involved. Thus, the fact that some of the neglected sources of uncertainty might be large compared to what is gained by our exact mathematical treatment does not qualify our approach as overly precise. The approach rather demonstrates the directions for future efforts into quantitative uncertainty estimation. As discussed above, our current formulation of the method may indeed be extended beyond the simplifications made. But as mathematical complexity increases in such case, extended formulations should be tailored to specific applications. In this sense, our formulation provides a minimal basis for the development of future uncertainty estimation methods.

Furthermore, the timescale-dependent uncertainties obtained from our method depend explicitly on assumptions regarding the structure of the supposed true climate signal $X(t) + Y(t)$, although this climate signal is the unknown quantity to be reconstructed from the proxy record. However, it is an inevitable fact that the timescale-dependent reconstruction uncertainties do actually depend on this structure, a fact that is made obvious by our method, and likewise by Amrhein (2019). One possible approach towards solving this problem would be an iterative procedure. (i) Assume a specific structure for the supposed true climate signal. (ii) Apply our method to obtain reconstruction uncertainties for a given proxy record. (iii) Check whether the reconstructed signal is consistent, under the obtained uncertainties, with the assumed structure, given its spectral or auto-correlation properties. (iv) If this is not the case, update the assumptions and repeat these steps.

Finally, although our method provides an advancement in the quantification of reconstruction uncertainties, it also introduces a number of model parameters which are associated with their own uncertainty. However, if we are to improve quantitative uncertainty estimates, our reconstruction uncertainty model helps to identify those parameters which are most important and, therefore, need to be determined at higher precision. For example, how much seasonality is imposed on a certain proxy at a given geographical location within a specific local ecological system? On the other hand, it is possible to investigate how parameter uncertainties translate into reconstruction uncertainties, as was shown for the seasonal phase parameter $\phi_c$. Nonethe-



less, the eventual benefit of uncertainty estimation methods like the one presented in this study, and of extensions based thereon, has still to be worked out in the future by systematically applying such methods to real data.

## 6    Conclusions

The present study introduces a method, the so-called Proxy Spectral Error Model (PSEM; see also Part II of this study by Dolman et al., 2019), for estimating timescale-dependent uncertainties of paleoclimate reconstructions obtained from single sediment proxy records. The method is based on an uncertainty model that takes into account proxy seasonality (together with orbital variations of the seasonal cycle amplitude), bioturbation, archive sampling parameters, and the effects of measuring only a finite number of signal carriers. For this model analytic expressions are derived for the power spectrum of the reconstruction

error, from which timescale-dependent reconstruction uncertainties can be obtained. This approach is motivated by the fact that the spectral structure of the error is equivalent to its auto-correlation structure which, in turn, determines how archive smoothing, sampling and averaging timescales affect the uncertainties. Various timescale-dependent uncertainty metrics can be defined and then be computed from the error power spectrum, by multiplying the spectrum by specific transfer functions and then integrating. This corresponds, in the time domain, to additional postprocessing steps performed on the reconstructed

time series. For example, it is possible to investigate the uncertainty reduction achieved by a lowpass filter with a given cut-off timescale, or to quantify the uncertainty of the difference between two time averages with given averaging timescales.

The method proves useful in different ways. First, it can serve to obtain quantitative uncertainty estimates for practical applications in paleoclimate science. This is demonstrated in Part II of this study (Dolman et al., 2019) where a number of application examples are presented. Second, the derived analytic expressions can be used to acquire a better qualitative

understanding of the structure of the uncertainties. In particular, we can conclude that

–   the reconstruction uncertainties can be decomposed into two components: (i) a component, the variance of which is obtained by multiplying the power spectrum of the supposed true climate signal by a transfer function and then integrating. This so-called error transfer function has a structure corresponding to a bandpass filter with its cut-off timescales given by the longest applied archive smoothing timescale and by a suitably chosen reference smoothing timescale (by analogy

with the transfer function discussed by Amrhein, 2019). Thus, multiplying the spectrum by the error transfer function corresponds to applying that bandpass filter to the supposed true climate signal. (ii) A white noise component that scales inversely with the number of signal carriers retrieved from each slice of sediment (and being subject to the same single laboratory measurement). Thus, in the asymptotic limit of infinitely many signal carriers this component vanishes. In the opposite limit, with only a single signal carrier being measured from each slice, the variance of this component

equals the variance that is contained in the supposed true climate signal at timescales shorter than the longest applied archive smoothing timescale. This component corresponds to what is referred to, by Dolman and Laepple (2018), as the noise created by aliasing of variability from inter- and intra-annual timescales. Depending on geographical location and climatic conditions, this white noise uncertainty component may be dominated by ENSO variability or by the seasonal cycle, for example.



- in the presence of proxy seasonality, such that the climate signal is recorded by the proxy only during a limited seasonal window each year, the abovementioned error transfer function has additional high-frequency peaks at the seasonal cycle frequency and its higher harmonics and, thus, corresponds to a multi-bandpass filter in this case. In consequence of this, a certain amount of variance is reallocated from the above white noise uncertainty component to the first component, although it appears there at the lowest frequencies because of spectral aliasing. Thus, proxy seasonality may generate uncertainties that are highly correlated in the time domain. In most cases this low-frequency uncertainty will be dominated by the seasonal cycle and its amplitude modulation caused by orbital variations (as demonstrated by Huybers and Wunsch, 2003, for example). Nonetheless, if the stochastic climate variablity is only weakly red such that it is associated with notable power near the seasonal cycle frequency, it may also give rise to low-frequency uncertainties, in particular, if the seasonal cycle is weak by comparison.

- if, in addition, the proxy abundance window is known to have a preferred seasonal timing throughout the year, then the contribution that the seasonal cycle signal (with its deterministic phase) makes to both of the above two uncertainty components is further modified. The white noise component can be larger or smaller than for random seasonal timing and, in particular, the first uncertainty component may include a (potentially time-varying) deterministic bias in this case. Moreover, the sum of their variances may change because of the in-phase subsampling from a deterministic signal.

- uncertainties caused by laboratory measurement errors are independent of the above components and, thus, the associated power spectral density can simply be added to the error power spectrum obtained from our method. In practice this uncertainty component is assumed to be white noise, such that it scales inversely with any averaging timescale.

Another interesting and future application of the derived analytic expressions would be the inference of the power spectrum of the true climate signal. Specifically, by setting the reference climate in our method to zero, and then repeating the entire derivation, one obtains the analytic expressions for the power spectrum of the climate reconstruction itself, rather than of its error. Thus, one obtains an operator that transforms the power spectrum of the supposed true climate signal into a spectrum subject to the distortions caused by the processes included in our reconstruction uncertainty model. Then, given all of the parameters of the uncertainty model, and assuming a parametric form for the true climate signal, it might be possible to estimate its parameters by means of a maximum likelihood approach (that investigates the likelihood, under a given set of parameters, of the power spectrum estimated from a specific proxy record). This essentially amounts to inverting the aforementioned operator, similar to the correction technique used by Laepple and Huybers (2013) that is motivated by the anti-aliasing approach of Kirchner (2005).





## Appendix A: Non-stationary variance of the white noise component $B_n^{(j)}$

The variance of $B_n^{(j)}$ is given by its auto-covariance function at lag zero, $\mathrm{Cov}_{(\epsilon)}(B_n^{(j)\star}, B_{n'}^{(j)})|_{n=n'}$. By substitution from (72)
it can be shown, after some algebraic transformations, that

$$\mathrm{Cov}_{(\epsilon)}\left(B_n^{(j)\star}, B_{n'}^{(j)}\right)\bigg|_{n=n'} = \mathcal{R} + \mathcal{R}^\star, \tag{A1}$$

with

$$\mathcal{R} = \alpha_c^2(R_1 + R_2) + 4\alpha_c^2\alpha_a(R_3 + R_4) + 2\alpha_c^2\alpha_a^2(R_5 + R_6 + R_7 + R_8), \tag{A2}$$

where

$$R_1 = D_1, \qquad R_2 = e^{-i2\phi_c}D_2, \qquad R_3 = e^{i(2\pi\nu_a t_n + \phi_a)}D_3, \qquad R_4 = \cos(2\phi_c)e^{i(2\pi\nu_a t_n + \phi_a)}D_4,$$

$$R_5 = D_5, \qquad R_6 = \cos(2\phi_c)D_6, \qquad R_7 = e^{i2(2\pi\nu_a t_n + \phi_a)}D_7, \qquad R_8 = \cos(2\phi_c)e^{i2(2\pi\nu_a t_n + \phi_a)}D_8, \tag{A3}$$

and the characteristic function differences $D_1$ to $D_8$ are given by, also using definition (32),

$$D_1 = C_{n,n'}(-\nu_c, \nu_c)|_{n=n'} - C_{l,l'}(-\nu_c, \nu_c)|_{l \neq l'}$$
$$= 1 - \mathrm{sinc}^2(\nu_c \tau_p), \tag{A4}$$

$$D_2 = C_{n,n'}(-\nu_c, -\nu_c)|_{n=n'} - C_{l,l'}(-\nu_c, -\nu_c)|_{l \neq l'}$$
$$= \mathrm{sinc}(2\nu_c \tau_p) - \mathrm{sinc}^2(\nu_c \tau_p), \tag{A5}$$

$$D_3 = C_{n,n'}(-\nu_c + \nu_a, \nu_c)|_{n=n'} - C_{l,l'}(-\nu_c + \nu_a, \nu_c)|_{l \neq l'}$$
$$= \hat{f}_{bs}^\star(\nu_a)\left[1 - \mathrm{sinc}^2(\nu_c \tau_p)\right], \tag{A6}$$

$$D_4 = C_{n,n'}(-\nu_c + \nu_a, -\nu_c)|_{n=n'} - C_{l,l'}(-\nu_c + \nu_a, -\nu_c)|_{l \neq l'}$$
$$= \hat{f}_{bs}^\star(\nu_a)\left[\mathrm{sinc}(2\nu_c \tau_p) - \mathrm{sinc}^2(\nu_c \tau_p)\right], \tag{A7}$$

$$D_5 = C_{n,n'}(-\nu_c - \nu_a, \nu_c + \nu_a)|_{n=n'} - C_{l,l'}(-\nu_c - \nu_a, \nu_c + \nu_a)|_{l \neq l'}$$
$$= 1 - |\hat{f}_{bs}^\star(\nu_a)|^2 \mathrm{sinc}^2(\nu_c \tau_p), \tag{A8}$$

$$D_6 = C_{n,n'}(-\nu_c - \nu_a, -\nu_c + \nu_a)|_{n=n'} - C_{l,l'}(-\nu_c - \nu_a, -\nu_c + \nu_a)|_{l \neq l'}$$
$$= \mathrm{sinc}(2\nu_c \tau_p) - |\hat{f}_{bs}^\star(\nu_a)|^2 \mathrm{sinc}^2(\nu_c \tau_p), \tag{A9}$$

$$D_7 = C_{n,n'}(-\nu_c + \nu_a, \nu_c + \nu_a)|_{n=n'} - C_{l,l'}(-\nu_c + \nu_a, \nu_c + \nu_a)|_{l \neq l'}$$
$$= \hat{f}_{bs}^\star(2\nu_a) - \hat{f}_{bs}^{\star 2}(\nu_a) \mathrm{sinc}^2(\nu_c \tau_p), \tag{A10}$$





$$D_8 = C_{n,n'}(-\nu_c + \nu_a, -\nu_c + \nu_a)|_{n=n'} - C_{l,l'}(-\nu_c + \nu_a, -\nu_c + \nu_a)|_{l \neq l'}$$

$$= \hat{f}_{bs}^{\star}(2\nu_a)\operatorname{sinc}(2\nu_c\tau_p) - \hat{f}_{bs}^{\star 2}(\nu_a)\operatorname{sinc}^2(\nu_c\tau_p). \tag{A11}$$

Since the auto-covariance contributions $R_3$, $R_4$, $R_7$ and $R_8$ depend on $t_n$, the variance of $B_n^{(j)}$ is non-stationary. Furthermore, it turns out that with $n \neq n'$ the characteristic function differences $D_1$ to $D_8$ are all zero and, thus, the auto-covariance con-
tributions $R_1$ to $R_8$ are all zero. This implies that the auto-covariance function of $B_n^{(j)}$ is non-zero only at lag zero ($n = n'$) and zero at all other lags ($n \neq n'$). Hence, $B_n^{(j)}$ is a white noise process, and from its definition (72) it follows that it has zero mean. Note, that (A4) is identical to (48) in section 3.1, but with $\nu = \nu_c$, and so the above procedure follows the same key idea (according to the approach of Balakrishnan, 1962) to extrapolate the auto-covariance function from non-zero lags towards lag zero.

From the above expressions, we can write the variance of $B_n^{(j)}$ as

$$\operatorname{Var}_{(\epsilon)}\left(B_n^{(j)}\right) = \mathcal{V}_B^{(0)} + \mathcal{V}_{B,n}^{(1)}\cos(2\pi\nu_a t_n + \phi_a + \phi_{b1})$$

$$+ \mathcal{V}_{B,n}^{(2')}\cos(4\pi\nu_a t_n + 2\phi_a + \phi_{b2}) + \mathcal{V}_{B,n}^{(2'')}\cos(4\pi\nu_a t_n + 2\phi_a + \phi_{b1}), \tag{A12}$$

with the amplitude of the stationary variance component,

$$\mathcal{V}_B^{(0)} = \sigma_c^2\left\{1 - \operatorname{sinc}^2(\nu_c\tau_p) + \cos(2\phi_c)\left[\operatorname{sinc}(2\nu_c\tau_p) - \operatorname{sinc}^2(\nu_c\tau_p)\right]\right\}$$

$$+ \sigma_c^2\sigma_a^2\left\{1 - M_{b1}^2\operatorname{sinc}^2(\nu_a\tau_s)\operatorname{sinc}^2(\nu_c\tau_p) + \cos(2\phi_c)\left[\operatorname{sinc}(2\nu_c\tau_p) - M_{b1}^2\operatorname{sinc}^2(\nu_a\tau_s)\operatorname{sinc}^2(\nu_c\tau_p)\right]\right\}, \tag{A13}$$

the amplitude of the variance component oscillating at frequency $\nu_a$,

$$\mathcal{V}_{B,n}^{(1)} = 2\sigma_c^2\sigma_a\sqrt{2}\left\{M_{b1}\operatorname{sinc}(\nu_a\tau_s)\left[1 - \operatorname{sinc}^2(\nu_c\tau_p)\right] + \cos(2\phi_c)M_{b1}\operatorname{sinc}(\nu_a\tau_s)\left[\operatorname{sinc}(2\nu_c\tau_p) - \operatorname{sinc}^2(\nu_c\tau_p)\right]\right\}, \tag{A14}$$

and the amplitudes of the variance components oscillating at frequency $2\nu_a$,

$$\mathcal{V}_{B,n}^{(2')} = \sigma_c^2\sigma_a^2\left\{M_{b2}\operatorname{sinc}(2\nu_a\tau_s)\left[1 + \cos(2\phi_c)\operatorname{sinc}(2\nu_c\tau_p)\right]\right\}, \tag{A15}$$

$$\mathcal{V}_{B,n}^{(2'')} = -\sigma_c^2\sigma_a^2\left\{M_{b1}^2\operatorname{sinc}^2(\nu_a\tau_s)\operatorname{sinc}^2(\nu_c\tau_p)\left[1 + \cos(2\phi_c)\right]\right\}, \tag{A16}$$

and where

$$\phi_{b2} = \arg\left[\hat{f}_b^{\star}(2\nu_a)\right] = 4\pi\nu_a\tau_b - \arctan(4\pi\nu_a\tau_b) \tag{A17}$$

and

$$M_{b2} = |\hat{f}_b^{\star}(2\nu_a)| = \left[1 + (4\pi\nu_a\tau_b)^2\right]^{-1/2}, \tag{A18}$$





and $\phi_{b1}$ and $M_{b1}$ are defined by (76) and (77), respectively. The time average of this variance over an infinitely long time
interval is then given by $\mathcal{V}_B^{(0)}$, provided that $\Delta t$ is not a multiple of $\nu_a^{-1}$. If the time average is taken over a finite time interval
of length $T$, centered at $t = 0$, the time mean variance is given by

$$\overline{\mathrm{Var}}_{(\epsilon)}\left(B_n^{(j)}\right) = \mathcal{V}_B^{(0)} + \mathcal{V}_{B,n}^{(1)}\cos(\phi_a + \phi_{b1})\operatorname{sinc}(\nu_a T)$$

$$+ \mathcal{V}_{B,n}^{(2')}\cos(2\phi_a + \phi_{b2})\operatorname{sinc}(2\nu_a T) + \mathcal{V}_{B,n}^{(2'')}\cos(2\phi_a + \phi_{b1})\operatorname{sinc}(2\nu_a T), \tag{A19}$$

provided that $\Delta t \ll \nu_a^{-1}$.

*Author contributions.* TL developed the underlying idea of the Proxy Spectral Error Model. TK, AMD and TL designed the conceptual
outline of the research. TK laid out and performed the mathematical derivation of the analytic expressions and wrote the manuscript, based
on numerous discussions with AMD and TL.

*Competing interests.* The authors declare that they have no conflict of interest.

*Acknowledgements.* This is a contribution to the SPACE ERC project; this project has received funding from the European Research Council
(ERC) under the European Union's Horizon 2020 research and innovation programme (grant agreement no. 716092). AMD was supported by
the German Federal Ministry of Education and Research (BMBF) through the PALMOD project (FKZ: 01LP1509C). The work profited from
discussions at the CVAS working group of the Past Global Changes (PAGES) programme. We also acknowledge the input from Christian
Proistosescu from University of Washington, for bringing to our awareness the formalism to describe the effects of jittered sampling in signal
processing.





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
