# Peer review of "Estimating the timescale-dependent uncertainty of paleoclimate records—a spectral approach. Part I: Theoretical concept"

_Climate of the Past, 2019_

## Referee Comment (RC1) · Anonymous Referee #1 · 13 Mar 2020

Since I am not necessarily familiar with this field, I cannot evaluate scientific significance. However, as far as I have read, this paper is mathematically rigorous and well-written. I have not yet completely checked the entire manuscript. Since the deadline has come, I am listing some minor points which I've noticed for now. Maybe I will add some comments later.

1. 3rd paragraph in Section 1: Scale-dependent correlations could be treated by the Gaussian process model (e.g., Rasmussen and Williams, 2006), which is also known as the kriging model in spatial statistics. It might be helpful to compare the proposed spectral approach with the Gaussian process approach if possible.

[Figure]

2. It would be convenient if the definition of the function $f_s$ is displayed on a separate line because this function is referred to later.

3. The definition of the operator '$*$', which is used in Eq. (4), is missing, although I understand it normally denotes convolution.

4. The meaning of the superscript $(j)$ in Eq. (5) is not clarified until Eq. (9). It should explicitly be explained around Eq. (5).

5. If I understand correctly, $p(\varepsilon)$ can also be written as

$$p(\varepsilon) = \frac{1}{\tau_p \nu_c} \sum_{k=-\infty}^{\infty} \Pi(\varepsilon - k\nu_c^{-1}; \tau_p)$$

and this form would be helpful to understand the sentence from L. 178 to L. 180. By the way, it seems to me the statement in L. 178 is not strict. In my understanding, $p(\nu_c^{-1}/2) = 0$ if $\tau_p < \nu_c^{-1}$ and $p(\nu_c^{-1}/2) = 2$ if $\tau_p = \nu_c^{-1}$. This might be fixed by modifying the definition of $\Pi$ in Eq. (3).

6. The shape of the PDF in Eq.(18) seems to be quite unnatural. It might be worth considering to use another widely used PDF for cyclic variables such as the von Mises distribution if possible.

7. I do not understand what the authors mean in the sentence from L. 338–340, and I cannot follow why $C(\nu = 0) = 1$ holds.

8. L. 356–357: I would suggest this sentence should be written in an equation, and I think it could be used for deriving Eq. (49). I cannot take how Eq. (48) is used for obtaining Eq. (49).

9. Around Eq. (50), it should be recalled what $F_{X,n}$ and $W_{X,n}$ mean. It is hard to find their meaning described in Page 10 when reading Page 16.

10. I do not understand how Eqs. (11) and (12) yield Eq. (53).

11. I think Eqs. (31) and (52) are also required for obtaining Eq. (55).

12. It would be helpful to display Eq. (1) again at the beginning of Section 3.2.

13. L. 431: The definition of $Y_n^{(j)}$ is not given.

14. It would be helpful if it is explained in detail how the parameters in Table 1 are chosen. I wonder whether the parameters can be estimated on the basis of some criterion such as the cross validation error or they are given according to some standard choice. I also wonder how sensitively results can be affected by the uncertainty of the parameters.

**References**

Rasmussen, C. E. and Williams, C. K. I.: Gaussian processes for machine learning, the MIT Press, Cambridge, Massachusetts, 2006.

---

## Referee Comment (RC2) · Anonymous Referee #2 · 31 Mar 2020

1) General comments

The manuscript addresses the estimation of the uncertainty of proxy-based reconstructions taking into account that uncertainties are potentially serially correlated. I think that the study of the uncertainties of climate proxy reconstructions is extremely relevant for the investigation of the climate of the past.

The approach described in the manuscript does not take into account calibration errors and dating uncertainties, which seems to be an obvious limitation of the method. However the option of preferring analytic solutions rather than more complex formulations that could easily become untreatable is somewhat justified in the text (section 5).

In my opinion the manuscript is solid and scientifically sound, and I only have some minor corrections to suggest.

2) Specific comments

- page 2, line 55: maybe it would be useful and illustrative to compare the results of numerical simulations with the results obtained using the analytic approach described in the manuscript.

- page 3, line 3: "representation of smoothing by bioturbation", please rephrase / clarify.

- Figure 1: since the stochastic signal and the deterministic seasonal signal are discussed first than the archive formation and sampling effects, maybe move Fig. 1a) to the last graph.

- line 214: maybe change the title of section 2.5 to "Reconstruction uncertainty", the term "versus" doesn't seem to be the most accurate here

3) Typos

- line 22 & line 37: typo (indispensible)

- line 117: equivalently

---

## Author Comment (AC1) · 10 Apr 2020

**Reply to Reviewer #1**

of the manuscript

**Estimating the timescale-dependent uncertainty of paleoclimate records—a spectral approach. Part I: Theoretical concept**

by Torben Kunz, Andrew M. Dolman, and Thomas Laepple,
submitted to *Climate of the Past* (https://doi.org/10.5194/cp-2019-150).
* * *
In the following, the original comments by the reviewer are shown in black, our replies in blue, and citations from the manuscript are shown on a gray background with changes in red:

Since I am not necessarily familiar with this field, I cannot evaluate scientific significance. However, as far as I have read, this paper is mathematically rigorous and well-written. I have not yet completely checked the entire manuscript. Since the deadline has come, I am listing some minor points which I've noticed for now. Maybe I will add some comments later.

We appreciate the effort of the reviewer to critically evaluate the manuscript and, in particular, to provide helpful and detailed comments on the mathematical formulation.

1. 3rd paragraph in Section 1: Scale-dependent correlations could be treated by the Gaussian process model (e.g., Rasmussen and Williams, 2006), which is also known as the kriging model in spatial statistics. It might be helpful to compare the proposed spectral approach with the Gaussian process approach if possible.

   Our approach allows us to estimate the expected uncertainty of a climate reconstruction, as a function of (averaging-) timescale, assuming a certain spectral structure of the true climate signal and certain processes that distort the signal, including the sampling procedure of the sediment material. Alternatively, it allows us to find the optimal sampling strategy or the optimal geographical locations that minimize the expected uncertainty.

   To our understanding, the above comment suggests to compare our approach to address the problem in the frequency domain (i.e., in terms of the spectral structure) to an alternative approach that addresses the problem in the time domain (i.e., in terms of the

auto-correlation structure), using the Gaussian process model. However, it is not clear to us how exactly such an approach may allow us to achieve the same kind of uncertainty estimates as does our spectral approach. To us the Gaussian process model seems more applicable to the problem of generating actual climate reconstructions that account for a known/assumed auto-correlation structure in the errors, rather than for deriving the structure of the error itself. Therefore, we did not add any comparative discussion to the manuscript.

2. It would be convenient if the definition of the function $f_s$ is displayed on a separate line because this function is referred to later.

We set $f_s$ to a separate line. Although it is referred to only once five lines later in (4), this change makes it easier to identify $f_s$ backward from (4), which is referred to several times throughout the manuscript.

3. The definition of the operator $*$, which is used in Eq. (4), is missing, although I understand it normally denotes convolution.

Given that the operator $*$ is implicitly explained on line 158, and that it is standard notation, we decided to leave this unchanged.

4. The meaning of the superscript $(j)$ in Eq. (5) is not clarified until Eq. (9). It should explicitly be explained around Eq. (5).

We included a short explanation into the sentence after (5), as follows:

with $\epsilon_n^{(j)} \sim f_{bs}(\epsilon)$, where $\epsilon_n^{(j)}$ represents the sampling jitter and $f_{bs}(\epsilon)$ the jitter PDF. In the above terminology, $\epsilon_n^{(j)}$ represents the timing error of a single signal carrier (labelled $j$) retrieved from a slice centered at $t = t_n$.

5. If I understand correctly, $p(\epsilon)$ can also be written as

$$p(\epsilon) = \frac{1}{\tau_p \nu_c} \sum_{k=-\infty}^{\infty} \Pi(\epsilon - k\nu_c^{-1}; \tau_p) \tag{6}$$

and this form would be helpful to understand the sentence from L. 178 to L. 180. By the way, it seems to me the statement in L. 178 is not strict. In my understanding $p(\nu_c^{-1}/2) = 0$ if $\tau_p < \nu_c^{-1}$ and $p(\nu_c^{-1}/2) = 2$ if $\tau_p = \nu_c^{-1}$. This might be fixed by modifying the definition of $\Pi$ in Eq. (3).

We agree that it is useful to also express $p(\epsilon)$ in the above alternative form, because then the reader does not necessarily need to imagine the convolution operation, in order to understand the sentence from L. 178 to L. 180. Therefore, we added a second line to (6):

$$p(\epsilon) = (\tau_p \nu_c)^{-1} \Pi(\epsilon; \tau_p) * \mathrm{III}(\epsilon; \nu_c^{-1})$$

$$= (\tau_p \nu_c)^{-1} \sum_{k=-\infty}^{\infty} \Pi(\epsilon - k\nu_c^{-1}; \tau_p); \qquad (6)$$

We also agree that the definition of the rectangle function $\Pi$ in (3) was not precise in the sense that the equality in $|t| \leq \tau/2$ is required only on one side of the box. We corrected the definition accordingly:

$$\Pi(t; \tau) = \begin{cases} 1 & \text{if } \cancel{|t| \leq \tau/2} \;\; -\tau/2 < t \leq \tau/2 \\ 0 & \text{otherwise} \end{cases} . \qquad (3)$$

6. The shape of the PDF in Eq. (18) seems to be quite unnatural. It might be worth considering to use another widely used PDF for cyclic variables such as the von Mises distribution if possible.

Using a uniform PDF is certainly somewhat unnatural in the sense that the actual proxy abundance is not abruptly switched on or off during the seasonal cycle. Therefore, a gradually declining PDF, like the von Mises distribution or a wrapped normal distribution, could be more realistic. However, the uniform distribution has the advantage that its Fourier transform is simply given by the sinc-function, a fact that is used many times and in different contexts throughout the manuscript. Furthermore, it is clearly stated that our choice of a uniform distribution is made for reasons of simplicity.

However, the above comment of the reviewer brought our attention to another issue related to this PDF. Given that on L. 128 the domain of possible values of $\phi_c$ is explicitly specified as the interval $(-\pi, \pi]$, the definition of the PDF in (18) is not precise as it does not indicate it is a wrapped distribution. Also the domain of possible values of $\Delta_{\phi_c}$ was not specified correctly. Therefore, we rewrote the definition:

For simplicity, we choose the wrapped uniform PDF

$$f_{\phi_c}(\phi_c) = \sum_{k=-1}^{1} \Delta_{\phi_c}^{-1} \Pi(\phi_c - \langle \phi_c \rangle_{\phi_c} + 2\pi k; \Delta_{\phi_c}), \qquad \text{with} \quad -\pi < \phi_c \leq \pi, \qquad (18)$$

with the expected seasonal phase $-\pi < \langle \phi_c \rangle_{\phi_c} \leq \pi$, and the seasonal phase uncertainty $0 \leq \Delta_{\phi_c} \leq\ < 2\pi$.

7. I do not understand what the authors mean in the sentence from L. 338-340, and I cannot follow why $C(\nu = 0) = 1$ holds.

In those two sentences from L. 338-341, the proof is provided that the integral of the jitter PDF, as defined by (8), is indeed equal to one, as it has to be for a PDF. However, in its current form it is perhaps too compact and, thus, a bit difficult to follow. Therefore, we rewrote the proof, and also made it a separate paragraph:

~~Finally, note, that the requirement, $\tau_s$ be a multiple of 1 year (made in section 2.3), implies that each of the peaks with $k \neq 0$ has one of its zeros at $\nu = 0$ because of the sinc-function involved in (39). Thus, since $\hat{f}_{bs}^{\star}(\nu = 0) = 1$, it follows from (37) that $C(\nu = 0) = 1$ and, hence, from (35) with $\nu = 0$ that the jitter PDF $p(\epsilon)f_{bs}(\epsilon)$ does indeed integrate to unity.~~

The proof that the jitter PDF $p(\epsilon)f_{bs}(\epsilon)$, as defined by (8), does indeed integrate to unity is equivalent to showing that $C(0) = 1$, as can be seen from (35) with $\nu = 0$. To demonstrate this, we evaluate $C(0)$ using (37), noting (i) that the term with $k = 0$ is equal to one at $\nu = 0$, because $\hat{f}_{bs}^{\star}(0) = 1$, according to (39), and (ii) that the remaining terms with $k \neq 0$ are all equal to zero at $\nu = 0$, because $\hat{f}_{bs}^{\star}(k\nu_c) = 0$, since $\text{sinc}(k\nu_c\tau_s) = 0$ according to the requirement $\tau_s$ be a multiple of 1 year (see section 2.3).

While making the above changes, we realized that the definition of the sinc-function in (38) was not well-written. We rewrote this accordingly:

$$\text{sinc}[(\cdot)] = \begin{cases} 1 & \text{if } \nu = 0 \\ \sin[\pi(\cdot)]/[\pi(\cdot)] & \text{if } \nu \neq 0 \end{cases}, \tag{38}$$

$$\text{sinc}(x) = \begin{cases} 1 & \text{if } x = 0 \\ \sin(\pi x)/(\pi x) & \text{if } x \neq 0 \end{cases}, \tag{38}$$

8. L. 356-357: I would suggest this sentence should be written in an equation, and I think it could be used for deriving Eq. (49). I cannot take how Eq. (48) is used for obtaining Eq. (49).

It is completely right that the statement in the above sentence (L. 356-357), together with the statement in the following sentence, could be used to obtain (49). The reason why we do not do this, but instead suggest to take the alternative way, using (45), (46) and (48), is simply that this will later help the reader seeing the analogy between this case and the more complicated case of the deterministic signal. In particular, the sentence in the appendix on L. 812-814 is an attempt to make this analogy explicit, including a reference to (48). For this reason we prefer to keep our explanation of how (49) is obtained.

Nonetheless, we agree that in its current form the explanation, to simply use (45), (46) and (48), may be puzzling. The additional information needed here is, probably, that one has to use those equations with $n = n'$, and then substitute from (48) into (46). Then the integral in (46) reduces to $\int_{-\infty}^{\infty} S_X d\nu - \int_{-\infty}^{\infty} |C(\nu)|^2 S_X d\nu$. The first of these integrals in obviously the variance of $X$, and the second integral is, from (45), the variance of $U_n$. Thus, it follows (49). To make this explicit, we changed the wording before (49):

This separates the full variance into two components (indicated in the figure by the transition in color at lag zero), such that,  by setting $n = n'$ in (45), (46) and (48), and substituting from (48) into (46),

$$\mathrm{Var}_{(X,\epsilon)}(V_n^{(j)}) = \mathrm{Var}_{(X)}(X) - \mathrm{Var}_{(X)}(U_n). \tag{49}$$

9. Around Eq. (50), it should be recalled what $F_{X,n}$ and $W_{X,n}$ mean. It is hard to find their meaning described in Page 10 when reading Page 16.

We added a short explanation, to recall their meaning, and a reference to section 2.5, where they are mentioned for the first time:

With these properties of the above components $U_n$ and $V_n^{(j)}$, we can now rewrite the error component $E_{X,n}$, defined by (14), also using the $X$-component of (9), as

$$E_{X,n} = F_{X,n} + W_{X,n}, \tag{50}$$

with

$$F_{X,n} = U_n - \tilde{X}_n \tag{51}$$

and

$$W_{X,n} = \frac{1}{N} \sum_{j=1}^{N} V_n^{(j)}, \tag{52}$$

where $F_{X,n}$ and $W_{X,n}$ represent the components of $E_{X,n}$ explained in section 2.5, that is, a component obtained by filtering the signal $X$, and a white noise component, respectively.

10. I do not understand how Eqs. (11) and (12) yield Eq. (53).

We modified the wording around (53):

 By analogy with (27), a spectral representation of the $X$-component $\tilde{X}_n$ of the reference climate signal,  defined by (11) and (12), is given by

$$\tilde{X}_n = \int_{-\infty}^{\infty} e^{i2\pi\nu t_n} \operatorname{sinc}(\nu\tau_r) dZ(\nu), \tag{53}$$

also using the convolution theorem, and where the sinc-function represents the Fourier transform of the moving average window in (12). Then  the auto-covariance function of $F_{X,n}$ is obtained [ … ]

11. I think Eqs. (31) and (52) are also required for obtaining Eq. (55).

Indeed, (52) is also needed, but we assumed the reader may still have it in mind. Nonetheless, it is of advantage to explicitly mention it again at this point. We also agree that for the second step, from (55) to (56), one may use (31). However, following the same argument as for point 8. above, we prefer to use (46) and (48) here. To make all this more explicit, we changed the wording around (55) and (56) accordingly:

Finally, because $\epsilon_n^{(j)}$ is also white in terms of $j$, we have,  from (49) and (52),

$$\operatorname{Var}_{(X,\epsilon)}(W_{X,n}) = \big[\operatorname{Var}_{(X)}(X) - \operatorname{Var}_{(X)}(U_n)\big]/N \tag{55}$$

$$= \frac{1}{N} \int_{-\infty}^{\infty} \big[1 - |C(\nu)|^2\big] S_X(\nu) d\nu, \tag{56}$$

where the second step may be obtained directly, from (52), by substituting from (48) into (46) with $n = n'$.

12. It would be helpful to display Eq. (1) again at the beginning of Section 3.2.

Although it could be helpful to repeat (1) here, it would also be somewhat unconventional to repeat an identical equation in the same manuscript. Therefore, we leave this unchanged.

13. L. 431: The definition of $Y_n^{(j)}$ is not given.

Although the definition of $Y_n^{(j)}$ is actually given by (5), this is indeed very far away from this point of the manuscript. Therefore, we modified the wording on L. 431-432:

> Again following the approach of Balakrishnan (1962), and by analogy with section 3.1, we evaluate $\left\langle Y_n^{(j)\star} Y_{n'}^{(j)} \right\rangle_\epsilon$, where $Y_n^{(j)} = Y(t_n + \epsilon_n^{(j)})$, as defined by (5). However, this  is not the auto-covariance function in this case, because [ . . . ]

For consistency, we also added the reference to (5) on L. 299, where $X_n^{(j)}$ is used for the first time:

> [ . . . ] and the signal with jittered sampling, $X_n^{(j)} = X(t_n + \epsilon_n^{(j)})$, as defined by (5), can be expressed as (Moore and Thomson, 1991) [ . . . ]

14. It would be helpful if it is explained in detail how the parameters in Table 1 are chosen. I wonder whether the parameters can be estimated on the basis of some criterion such as the cross validation error or they are given according to some standard choice. I also wonder how sensitively results can be affected by the uncertainty of the parameters.

The issue of how specific parameter values may be chosen is discussed in detail in Part II of this study (Dolman et al., 2019). We included a sentence after the reference to Table 1 on L. 93:

> [ . . . ] including an explanation of the involved parameters. A complete list of the model parameters is provided by Table 1. For possible sources and specific choices of parameter values, see Part II of this study (Dolman et al., 2019) and, in particular, their Table 1. Note, that the reconstruction uncertainty model defined in this section [ . . . ]

---

## Author Comment (AC2) · 10 Apr 2020

**Reply to Reviewer #2**

of the manuscript

**Estimating the timescale-dependent uncertainty of paleoclimate records—a spectral approach. Part I: Theoretical concept**

by Torben Kunz,Andrew M. Dolman, and Thomas Laepple,
submitted to *Climate of the Past* (https://doi.org/10.5194/cp-2019-150).
* * *
In the following, the original comments by the reviewer are shown in black, our replies in blue, and citations from the manuscript are shown on a gray background with changes in red:

1) General comments

The manuscript addresses the estimation of the uncertainty of proxy-based reconstructions taking into account that uncertainties are potentially serially correlated. I think that the study of the uncertainties of climate proxy reconstructions is extremely relevant for the investigation of the climate of the past.

The approach described in the manuscript does not take into account calibration errors and dating uncertainties, which seems to be an obvious limitation of the method. However the option of preferring analytic solutions rather than more complex formulations that could easily become untreatable is somewhat justified in the text (section 5).

In my opinion the manuscript is solid and scientifically sound, and I only have some minor corrections to suggest.

We appreciate the helpful and constructive comments and suggestions of the reviewer. Below the specific comments are addressed point by point.

2) Specific comments

1. page 2, line 55: maybe it would be useful and illustrative to compare the results of numerical simulations with the results obtained using the analytic approach described in the manuscript.

(a) In order to verify our analytic results, we performed extensive sets of numerical simulations of exactly the same model as described by the analytic approach. By doing so we were able to bring analytic and numerical solutions into perfect agreement, with different settings of the various model parameters. This was an important step for achieving full confidence in our analytic results.

(b) Apart from this, one may conduct a more general comparison, in an application-oriented context, between (i) the reconstruction uncertainties obtained by the approach presented in this study, and (ii) the uncertainties obtained from alternative approaches like, for example, different proxy forward models or more simplistic approaches ignoring any serial correlation of reconstruction errors. However, such methodological comparisons should be made in a systematic manner and can be expected to become rather comprehensive, providing interesting material for future studies.

2. page 3, line 3: "representation of smoothing by bioturbation", please rephrase / clarify.

   We rephrased this to clarify the meaning of the sentence on lines 62-63:

    The fact that archive smoothing is represented by bioturbation limits the validity of the method in its current form [ . . . ]

3. Figure 1: since the stochastic signal and the deterministic seasonal signal are discussed first than the archive formation and sampling effects, maybe move Fig. 1a) to the last graph.

   It is true that the order of the panels in Fig. 1 differs from the order they are mentioned in the text, where they appear in the order (b), (c), (a), (d). However, first, it would not be meaningful to show the panels in this order because it makes more sense to have panels (b), (c) and (d) together, and, second, the PDF shown in (a) is, in some sense, a more abstract quantity than the timeseries, and like an operator that acts on what is shown beneath it in panels (b)-(d). Therefore, we leave this unchanged.

4. maybe change the title of section 2.5 to "Reconstruction uncertainty", the term "versus" doesn't seem to be the most accurate here

   We agree it is not the most intuitive section title. We changed it to make it consistent with the title of section 2.4:
* * *
   **2.5 Definition of reconstruction uncertainty**

3) Typos

1. line 22 & line 37: typo (indispensible)

   Corrected.

2. line 117: equaivalently

   Corrected.